# Text Generation as Continuous Latent Dynamics via Reinforcement Learning

**Chen Jia** [1]

## Abstract

We propose to model text generation as a continuous-time latent dynamical process, where token generation is formulated as a Markov decision process whose internal state evolves via a neural ODE. This formulation bridges discrete token sequences and continuous semantic evolution, providing a theoretically grounded framework for text generation with continuous-time latent states. The framework is optimized via reinforcement learning, maximizing a composite objective that integrates cumulative rewards with a Kullback–Leibler divergence regularization term from a pre-trained language model. Both theoretical and empirical results demonstrate that our **C**ontinuous-**T**ime **L**atent **L**anguage **M**odel (**CT-LLM**) achieves superior effectiveness and efficiency in text generation, establishing a new paradigm for continuous-time language modeling.

## 1. Introduction

Text generation has become a cornerstone of modern natural language processing (NLP), powering applications from conversational agents (Vinyals & Le, 2015; Ouyang et al., 2022) to creative writing assistants (Yang et al., 2022; 2023). To align large language models (LLMs) with human intentions, reinforcement learning from human feedback (RLHF) (Christiano et al., 2017; Stiennon et al., 2020; Ouyang et al., 2022; Bai et al., 2022) has proven effective in guiding models to produce outputs that better match human preferences. These approaches typically optimize a language policy to generate text that maximizes expected cumulative rewards, which serve as proxies for human preference.

Standard RLHF methods frame text generation as a Markov decision process (MDP), where the policy generates one token at a time conditioned on a state representing the previously generated tokens (Christiano et al., 2017; Stiennon

[1]Independent Researcher. Correspondence to: Chen Jia <jiachenwestlake@gmail.com>.

*Proceedings of the 43rd International Conference on Machine Learning*, Seoul, South Korea. PMLR 306, 2026. Copyright 2026 by the author(s).

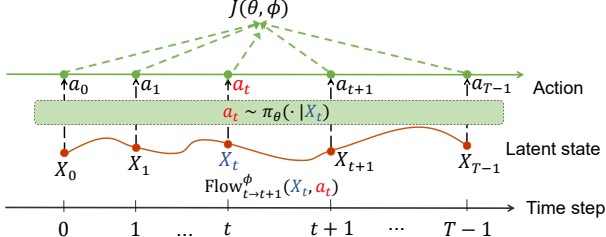

*Figure 1.* Overall Markov decision process (MDP) framework of CT-LLM, where a decision $a_t$ (which token to output) is made at each discrete time step $t \in \{0, 1, \ldots, T-1\}$. Between these steps, the latent state $X_t$ represents the continuous semantic manifold, modeling the flow of meaning between tokens through differential equations.

et al., 2020; Ouyang et al., 2022). In this conventional framework, the state is merely a "passive record" of the discrete token sequence, it captures which tokens occurred but discards rich information about the continuous trajectory of underlying meaning. The model has no access to the path between tokens, only to the discrete endpoints themselves. A promising yet underexplored alternative is to view text generation not as a sequence of discrete jumps, but as a smooth, continuous evolution through a hidden "meaning" space, analogous to how continuous flow models generate data (Tang et al., 2021).

In this paper, we propose a new framework that models text generation as a continuous, hidden-state process optimized with reinforcement learning, as illustrated in Figure 1. We formulate the task as an MDP where a discrete decision (which token to output) occurs at each step. Spefically, between these steps, the model's internal state does not remain static, it evolves smoothly and continuously according to an ordinary differential equation (ODE) within a compact latent space. This continuous "flow", parameterized by a neural network, enables the model's internal representation to change gradually and coherently.

To effectively train this continuous-discrete hybrid framework, we need to jointly optimize both the discrete decision-making process and the continuous latent dynamics. We introduce a joint optimization algorithm in which the policy (responsible for selecting the next action $a_t$) and the dynamics model (governing the state evolution $X_s$, $s \in (t, t+1]$ are trained together via stochastic gradient ascent (SGA)

to maximize a combined objective: achieving high cumulative reward (e.g., from a task-specific reward model) for the generated sequence through joint policy and latent ODE optimization. Additionally, we incorporate a Kullback–Leibler (KL) divergence regularization term to ensure training stability and to keep the model's behavior close to that of a strong pre-trained language model.

Our main contributions are:

- **A continuous-time latent dynamics framework:** We introduce a new way to think about text generation, where the core process is a smooth, continuous evolution of a hidden state governed by a neural ODE. This provides a solid foundation for generating more coherent and fluent text.

- **An efficient RL training algorithm:** We create a practical training objective that blends cumulative rewards with policy regularization. Our algorithm efficiently computes gradients for the ODE using the adjoint sensitivity method and combines them with policy gradient techniques for stable training.

- **Theoretical and empirical validation:** We prove that our algorithm converges at a standard rate of $\tilde{\mathcal{O}}(1/\sqrt{K})$ for $K$ optimization iterations, as well as a suboptimality bound for KL-regularized training objective. In experiments on standard benchmarks, our method, **CT-LLM**, outperforms strong discrete-time baselines.

**Conflict of Interest Disclosure.** The authors declare no competing financial interests or personal relationships that could have appeared to influence the work reported in this paper.

## 2. Related Work

**Language model Fine-tuning with reinforcement learning.** The use of reinforcement learning to align language models with human preferences has become a dominant paradigm (Ranzato et al., 2016; Li et al., 2016). Seminal work on Reinforcement Learning from Human Feedback (RLHF) (Christiano et al., 2017; Stiennon et al., 2020; Ouyang et al., 2022; Bai et al., 2022) employed reward models learned from preference data and fine-tuned policies via algorithms like Proximal Policy Optimization (PPO) (Schulman et al., 2017), achieving significant improvements in dialogue and instruction following. Subsequent efforts simplified the pipeline, notably Direct Preference Optimization (DPO) and its variants (Rafailov et al., 2023; Azar et al., 2024; Ethayarajh et al., 2024; Meng et al., 2024; Hong et al., 2024), which bypasses explicit reward modeling by directly optimizing a policy on preference pairs.

Concurrently, knowledge distillation techniques have been used to transfer capabilities from large "teacher" models to smaller "student" models, with on-policy variants showing effectiveness for language model compression (Lin et al., 2020; Agarwal et al., 2024; Gu et al., 2024; Wen et al., 2023; Jia, 2024). Despite their power, these methods fundamentally rely on discrete-time state representations, typically employing Transformer architectures with autoregressive generation loops. Our work departs from this by introducing a continuous-time latent state process, offering a different inductive bias for modeling token sequences and their evolution.

**Continuous-time and latent dynamical models for sequences.** Modeling sequential data with continuous-time dynamics has a rich history (Funahashi & Nakamura, 1993; Bailer-Jones et al., 1998). Neural Ordinary Differential Equations (Neural ODEs) (Chen et al., 2018) demonstrated how parameterized ODEs can model the continuous transformation of hidden states, inspiring applications in time-series modeling and irregularly sampled data. In natural language processing (NLP), latent variable models and state-space models have been explored to capture deeper dependencies (Bowman et al., 2016; Ziegler & Rush, 2019; Gu et al., 2021). For example, models leveraging ODEs or Stochastic Differential Equations (SDEs) have been proposed for text representation learning or controlled generation (Chen et al., 2018; Rubanova et al., 2019; Kidger et al., 2020; Song et al., 2021; Kidger et al., 2021; Li et al., 2022; Kan et al., 2026). However, these approaches have primarily focused on learning representations or interpolating between sequences, rather than directly governing the token-by-token generative process of a language model policy. Our key innovation is to formulate the core text generation mechanism, specifically the progression of the generator's internal state between tokens, as a learnable continuous-time dynamical system tightly coupled with a policy that emits discrete tokens from this evolving state.

## 3. Formulation and Preliminaries

**Text generation as MDP.** We follow a classical Markov Decision Process (MDP) formulation for discrete-time text generation. Given an input *prompt* $x \in \mathcal{X}$ drawn from the input distribution $\mu$, we define the corresponding natural language output as a sequence $(y_1, y_2, \ldots, y_T)$, where each token $y_t$ corresponds to an *action* $a_t = y_{t+1} \in \mathcal{A}$ for each time step $t \in \{0, \ldots, T-1\}$, which only depends on the current *state* $s_t \in \mathcal{S}$ for $t \geq 0$. Let $s_0$ be the initial state obtained by encoding the input $x$. The probability for a trajectory $\tau = (s_0, a_0, \ldots, s_{T-1}, a_{T-1}, s_T)$ given the initial state $s_0$ is represented by: $\mathbb{P}(\tau \mid s_0) = \prod_{t=0}^{T-1} \mathbb{P}(a_t \mid s_t)\mathbb{T}(s_{t+1} \mid s_t, a_t)$, where $\mathbb{T}(\cdot \mid s_t, a_t) \in \Delta_{|\mathcal{S}|}$ denotes the transition kernel.

**Latent dynamical modeling of text generation.** We further hypothesize that the state is a continuous-time process such that the state $X_s := X(s)$ is defined on the real interval $s \in [0, T]$. Let $\pi(\cdot \mid X) \in \mathcal{P}(\mathcal{A})$ be a language policy given a latent state $X \in \mathbb{R}^d$. Consider a latent state process $\{X_s\}_{0 \leq s \leq T}$ governed by a ordinary differential equation (ODE):

$$dX_s = b(s, X_s, a_t)ds, \quad s \in (t, t+1], \qquad (1)$$

where $b : \mathbb{R}_{\geq 0} \times \mathbb{R}^d \times \mathcal{A} \to \mathbb{R}^d$ is a given function and the initial state $s_0 = X_0$. Given the state $s_t := X_t$ on each time step $t \in \{0, \dots, T-1\}$, the action $a_t$ for $t \in \{0, \dots, T-1\}$ is drawn from a policy: $a_t \sim \pi(\cdot \mid X_t)$. Given the latent state and action pair $(X_t, a_t)$, the transition kernel $\mathbb{T}(\cdot \mid X_t, a_t)$ is deterministic and the latent state at the next step $s_{t+1} := X_{t+1}$ can be represented as a continuous flow: $X_{t+1} = \text{Flow}_{t \to t+1}(X_t, a_t)$ by OED in Eq. (1).

**Performance metric.** We denote $(X_s^\pi, a_t^\pi)$ where $s \in (t, t+1]$ and $t \in \{0, 1, \dots, T-1\}$, as the state and action pair by the stochastic dynamics Eq. (1) under the policy $\{\pi(\cdot \mid x) : x \in \mathbb{R}^d\}$. Our goal is to find the optimal policy $\pi^*$ that maximizes the discounted cumulative reward over a finite time horizon in expectation, which is defined as the optimal value function $V^*$ over the input distribution $\mu$ as follows.

$$V^*(\mu) = \max_\pi \mathbb{E}_{x \sim \mu} \mathbb{E}_\pi \left[ \sum_{t=0}^{T-1} \gamma^t r(X_t^\pi, a_t^\pi) \mid X_0^x \right], \quad (2)$$

where $r : \mathbb{R}^d \times \mathcal{A} \to \mathbb{R}$ denotes the reward function, $X_0^x$ denotes the initial state obtained by encoding the input $x$, $\gamma$ denotes the discount factor. The value function reduces to a deterministic function:

$$V^\pi(t, X_t^\pi) = \mathbb{E}_\pi \left[ \sum_{t'=t}^{T-1} \gamma^{t'-t} r(X_{t'}^\pi, a_{t'}^\pi) \mid X_t^\pi \right]. \quad (3)$$

The performance metric is the expected value over the input: $V^\pi(\mu) := \mathbb{E}_{x \sim \mu}[V^\pi(0, X_0^x)]$. Our goal is to find the optimal policy $\pi^*$ that maximizes the expected value $V^* := \max_\pi V^\pi(\mu)$. We can define the Q-function as $Q^\pi(t, X_t^\pi, a_t^\pi) = \mathbb{E} \left[ r(X_t^\pi, a_t^\pi) + \gamma V^\pi(t+1, X_{t+1}^\pi) \right]$ and define the advantage function as $A^\pi(t, X_t^\pi, a_t^\pi) := Q^\pi(t, X_t^\pi, a_t^\pi) - V^\pi(t, X_t^\pi)$.

## 4. RL for Continuous Latent Text Generation

The continuous latent formulation in the previous section introduces a dynamics model governed by an ODE, which provides a flexible and differentiable way to model token-level transitions. However, directly optimizing the policy in such a continuous-time latent space faces two main challenges: (i) the policy gradient needs to backpropagate through the

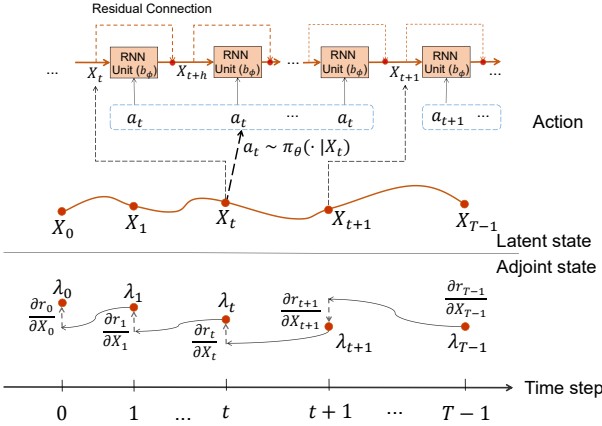

*Figure 2.* Forward and backward passes of ODE solver.

ODE solver, and (ii) the ODE dynamics by $b_\phi$ parameterized by $\phi$ and policy $\pi_\theta$ parameterized by $\theta$ are coupled through the latent trajectory $\{X_t^\theta\}_{t=0}^T$. To address these challenges, we propose a reinforcement learning framework that integrates the adjoint sensitivity method for efficient gradient computation, enabling end-to-end training of both the policy and the latent dynamics.

### 4.1. RL Formulation

**Objective function.** We construct $b_\phi$ with a neural network parameterized by $\phi$, such that the latent state $\{X_s^\theta\}_{0 \leq s \leq T}$ governed by the parameterized ODE of Eq. (1):

$$dX_s^\theta = b_\phi(s, X_s^\theta, a_t^\theta)ds, \quad s \in (t, t+1], \qquad (4)$$

where the action $a_t^\theta$ is drawn from a parameterized policy: $a_t^\theta \sim \pi_\theta(\cdot \mid X_t^\theta)$ for $t \in \{0, 1, \dots, T-1\}$. From the value function of Eq. (3), we denote the loss function w.r.t. $\theta$, $\phi$ as follows:

$$J(\theta, \phi) = \mathbb{E}_{x \sim \mu} \mathbb{E}_{\pi_\theta} \left[ \sum_{t=0}^{T-1} \gamma^t r(X_t^\theta, a_t^\theta) \mid X_0^x \right]. \quad (5)$$

**ODE solution and efficient gradient computation.** We construct $b_\phi$ with a neural network parameterized by $\phi$, which can be viewed as a recurrent neural network (RNN) unit. Following Neural ODE (Chen et al., 2018; Rubanova et al., 2019; Kidger et al., 2020), an approximate ODE solver can be seen as a RNN with residual connections between layers, as shown in Figure 2. Given a sample of action sequence $\{a_t^\theta\}_{t=0}^{T-1}$, for any $t \in \{0, 1, \dots, T-1\}$, set the discrete time step as $t_n = t + nh$ and using $X_n^\theta = X_{t_n}^\theta$ for short, the computation process using the Euler method can be formally represented as

$$X_{n+1}^\theta = X_n^\theta + b_\phi(t_n, X_n^\theta, a_t^\theta) h, \qquad (6)$$

where the step size is $h = 1/N$. We note that as the step size per token interval $N \to \infty$, the ODE solver convergence to the neural ODE in Eq. (1). We denote the ODE solution process as an operation ODESolve compute $X_{t+1}^\theta = \text{ODESolve}(X_t^\theta, b_\phi(\cdot, \cdot, a_t^\theta), t, t+1)$ for $t \in \{0, \ldots, T-1\}$.

We denote the empirical version of the objective in Eq. (5) as $\hat{J}(\theta, \phi)$ with a trajectory, which implicitly depends on $\phi$ through the latent trajectory $\{(X_t^\theta, a_t^\theta)\}_{t=0}^T$, where the latent states are governed by the parameterized ODE in Eq. (4). To compute an unbiased gradient estimator $\partial J/\partial \phi$, we adopt the continuous adjoint sensitivity method of Chen et al. (2018), which can be expressed as the following statements:

For each interval $(t, t+1]$, we introduce an adjoint state $\lambda_s \in \mathbb{R}^d$ defined as the total derivative of the objective with respect to the latent state at time $s$: $\lambda_s := \frac{\partial \hat{J}}{\partial X_s^\theta}, s \in (t, t+1]$.

The adjoint state satisfies the following backward-in-time ODE:

$$\frac{\mathrm{d}\lambda_s}{\mathrm{d}s} = -\left(\frac{\partial b_\phi}{\partial X_s^\theta}(s, X_s^\theta, a_t^\theta)\right)^\top \lambda_s, \quad s \in (t, t+1], \quad (7)$$

with terminal condition $\lambda_{T-1} = \mathbf{0}$. At each discrete time step $t = T-1, \ldots, 0$, the adjoint is updated to incorporate the immediate reward gradient: $\lambda_t \leftarrow \lambda_t + \gamma^t \frac{\partial r_t}{\partial X_t^\theta}$, where $r_t$ denotes the $t$-step reward.

The gradient with respect to the dynamics parameters $\phi$ is then given by the integral of the adjoint weighted by the sensitivity of the vector field:

$$\frac{\partial \hat{J}}{\partial \phi} = \sum_{t=0}^{T-1} \int_t^{t+1} \left(\frac{\partial b_\phi}{\partial \phi}(s, X_s^\theta, a_t^\theta)\right)^\top \lambda_s \, \mathrm{d}s.$$

This formulation enables memory-efficient backpropagation through the ODE solver without storing the full forward trajectory, and provides an unbiased estimator suitable for stochastic gradient-based optimization, i.e., $\mathbb{E}_{x \sim \mu} \mathbb{E}_{\pi_\theta} \left[\partial \hat{J}/\partial \phi \mid X_0^x\right] = \partial J/\partial \phi$.

**Policy gradient method.** We adopt an advantage-based policy gradient formulation to optimize the policy parameters $\theta$. The policy gradient (PG) is given by:

$$\nabla_\theta J(\theta, \phi)$$
$$= \mathbb{E}\left[\sum_{t=0}^{T-1} \gamma^t \nabla_\theta \log \pi_\theta(a_t^\theta \mid X_t^\theta) A^{\pi_\theta}(t, X_t^\theta, a_t^\theta)\right], \quad (8)$$

where the expectation is taken on the trajectory distribution induced by the input distribution $\mu$ and the policy $\pi_\theta$, $A^{\pi_\theta}(t, X_t^\theta, a_t^\theta)$ denotes the advantage function.

In practice, policy gradient methods estimate this gradient from sampled trajectories and update $\theta$ via stochastic

gradient ascent (SGA). When implemented with automatic differentiation frameworks, one constructs a surrogate objective whose gradient matches the policy gradient estimator. A widely used choice is the *REINFORCE* objective (Williams, 1992): Sampling a trajectory $x \sim \mu$, $\tau \sim \pi_\theta \mid X_0^x$, the *empirical* policy gradient is computed by:

$$\hat{\nabla}_\theta J(\theta, \phi) = \sum_{t=0}^{T-1} \gamma^t \nabla_\theta \log \pi_\theta(a_t^\theta \mid X_t^\theta) \hat{G}_t^{\pi_\theta}, \quad (9)$$

where $\hat{G}_t^{\pi_\theta} := \sum_{t'=t}^{T-1} \gamma^{t'-t} r(X_{t'}^\theta, a_{t'}^\theta)$ denotes the empirical policy gradient for each time step $t \in \{0, \ldots, T-1\}$. The policy gradient theorem (Sutton et al., 2018) gives that: $\mathbb{E}_{x \sim \mu} \mathbb{E}_{\pi_\theta} \left[\hat{\nabla}_\theta J(\theta, \phi) \mid X_0^x\right] = \nabla_\theta J(\theta, \phi)$, where the empirical PG in Eq. (9) is an unbiased estimator of PG in Eq. (8).

### 4.2. Optimization Algorithm

**Training procedure.** Algorithm 2 outlines the complete training procedure. In each training iteration, an initial latent state $X_0^x$ is obtained by encoding a sampled prompt $x$ via latent dynamics of Eq. (6). Given $X_0^x$, a sampling pass (Algorithm 1) alternates between action sampling from the policy $\pi_\theta(\cdot \mid X_t^\theta)$ and solving the neural ODE over the interval $(t, t+1]$ to generate the trajectory $\tau = (X_0^x, a_0^\theta, \ldots, X_{T-1}^\theta, a_{T-1}^\theta)$. Then, the policy parameters $\theta$ are updated via stochastic gradient ascent (SGA) with an empirical policy gradient $\hat{\nabla}_\theta J$ computed in Eq. (9) (Williams, 1992; Sutton et al., 2018). Then, to efficiently compute gradients w.r.t. the neural ODE parameters $\phi$, we employ the adjoint sensitivity method (Chen et al., 2018). This approach formulates gradient computation as an augmented ODE system that is integrated backward in time (Lines 8–12), avoiding the need to store intermediate states from the forward pass. Specifically, the adjoint state $\hat{\lambda}_t$ is initialized at the terminal time and updated recursively by incorporating reward gradients $\partial r_t/\partial \hat{X}_t$, while the *empirical* gradient $\hat{\nabla}_\phi J$ is accumulated through the backward integration of the augmented dynamics.

**Convergence analysis.** Our algorithm alternates between policy updates via a policy gradient method and dynamics updates via an adjoint sensitivity method on the ODE parameters. Before showing the convergence of this bilevel optimization scheme, we need the following assumptions:

**Assumption 4.1.** We make the following standard assumptions (detailed formulations are illustrated in Appendix A.1):

- **Boundedness:** $\|b_\phi\| \leq B$, $\|\partial b_\phi/\partial \phi\| \leq B_{\partial \phi}$, $\|\partial J/\partial X_t\| \leq B_{\partial X}$.

- **Lipschitzness:** $b_\phi(s, X, a_t)$ is $M_t$-Lipschitz in $s$ and $L_b$-Lipschitz in $X$; $\partial b_\phi/\partial \phi$ is $L_t$-Lipschitz in $s$ and

---

**Algorithm 1** Sampling Process for Action Sequence

---

1: **Input:** Initial latent state $X_0^x$
2: **Output:** Steps of actions $\{a_t^\theta\}_{t=0}^{T-1}$, latent state of the last step $X_{T-1}^\theta$
3: **for** $t = 0$ **to** $T - 1$ **do**
4:     Sample an action via the policy: $a_t^\theta \sim \pi_\theta(\cdot \mid X_t^\theta)$
5:     ODE solution: $X_{t+1}^\theta \leftarrow \text{ODESolve}(X_t^\theta, b_\phi(\cdot, \cdot, a_t^\theta), t, t+1)$ ▷ Iter. imple. Eq. (6) on $t_n$ from $n = 0$ to $n = N - 1$
6: **end for**

---

**Algorithm 2** Joint Training Procedure in RL

---

1: **Input:** Policy parameters $\theta$, neural ODE parameters $\phi$, reward model $r$, reference policy $\pi_{\text{ref}}$;
        Input dataset $\mathcal{D}$; Learning rates for policy $\eta_\theta$ and neural ODE $\eta_\phi$; discount factor $\gamma$
2: **repeat**
3:     Sample a prompt $x \in \mathcal{D}$ and encode it as an initial latent state $X_0^x$ via latent dynamics of Eq. (6)
4:     Sample action sequence $\{a_t^\theta\}_{t=0}^{T-1}$ by Algorithm 1 based on the initial latent state $X_0^x$
5:     Update the parameters of policy: $\theta \leftarrow \theta + \eta_\theta \hat{\nabla}_\theta J(\theta, \phi)$          ▷ SGA updating of policy parameters
6:     Sample a prompt $x \in \mathcal{D}$ and encode it as an initial state $X_0^x$ via latent dynamics of Eq. (6)
7:     Sample $\{a_t^\theta\}_{t=0}^{T-1}$ and $X_{T-1}^\theta$ by Algorithm 1 and initialize the adjoint states: $\hat{X}_{T-1}^\theta \leftarrow X_{T-1}^\theta, \hat{\lambda}_{T-1} \leftarrow \mathbf{0}, \hat{\nabla}_\phi J \leftarrow \mathbf{0}$
8:     **for** $t = T - 1$ **to** $1$ **do**
9:         Update the adjoint sensitivity $\hat{\lambda}_t \leftarrow \hat{\lambda}_t + \gamma^t \frac{\partial r_t}{\partial \hat{X}_t^\theta}$ and set the initial augmented state: $\bar{\lambda}_t = [\hat{X}_t^\theta, \hat{\lambda}_t, \hat{\nabla}_\phi J]$
10:         Define an augmented dynamics $\bar{b} : ([\hat{X}_s^\theta, \hat{\lambda}_s, \cdot], s, \phi) \mapsto [b_\phi(s, \hat{X}_s^\theta, a_t^\theta), -\hat{\lambda}_s^\top \frac{\partial b_\phi}{\partial \hat{X}_s^\theta}, -\hat{\lambda}_s^\top \frac{\partial b_\phi}{\partial \phi}]$
11:         $[\hat{X}_{t-1}^\theta, \hat{\lambda}_{t-1}, \hat{\nabla}_\phi J] \leftarrow \text{ODESolve}(\bar{\lambda}_t, \bar{b}, t, t-1)$          ▷ The ODE solution process similar to Eq. (6)
12:     **end for**
13:     Update the parameters of neural ODE: $\phi \leftarrow \phi + \eta_\phi \hat{\nabla}_\phi J(\theta, \phi)$          ▷ Maximize the objective w.r.t. $\phi$ via SGA
14: **until** converge (or do not reach the maximum iteration $K$) and **return** ODE dynamics $b_\phi$ and policy $\pi_\theta$

---

$L_{\partial \phi}$-Lipschitz in $X$; $\partial b_\phi / \partial X$ is $L_t'$-Lipschitz in $s$ and $L_{\partial X}$-Lipschitz in $X$; $\partial J / \partial \phi$ is $L_J$-Lipschitz in $\theta$.

- **Smoothness:** $J(\theta, \phi)$ is $L_\theta$-smooth in $\theta$ and $L_\phi$-smooth in $\phi$.

- **Bounded variance:** $\mathbb{E}\left[\|\hat{\nabla}_\theta J - \nabla_\theta J\|^2\right] \leq \sigma^2$, $\mathbb{E}\left[\|\partial \hat{J}/\partial \phi - \partial J/\partial \phi\|^2\right] \leq (\sigma')^2$.

**Theorem 4.2** (**Convergence to Stationary Point**). *Under the standard assumptions in Assumption 4.1 and assumption that the token interval step number satisfies $N \geq 2\eta_\theta T \tilde{C} / ((\eta_\phi - 3L_\phi \eta_\phi^2/2)\|\partial J(\theta, \phi)/\partial \phi\|)$ where $\tilde{C}$ denotes a constant (defined in Eq. (49)), we set the learning rates as $\eta_\theta = \frac{\alpha_\theta}{\sqrt{K}}$ and $\eta_\phi = \frac{\alpha_\phi}{\sqrt{K}}$ with some constant $\alpha_\theta, \alpha_\phi$ for $K$ optimization iterations. Then, the optimization algorithm satisfies:*

$$\min_{0 \leq k \leq K-1} \mathbb{E}\left[\|\nabla_{\theta,\phi} J(\theta_k, \phi_k)\|^2\right]$$

$$\leq \tilde{\mathcal{O}}\left(\frac{\kappa}{\sqrt{K}C'} + \frac{L_J^2 \alpha_\theta^2 \sigma^2}{K}\right),$$

*where $C' = \min\left\{\frac{1}{2}\left(\alpha_\phi - \frac{L_\phi \alpha_\phi}{2\sqrt{K}}\right), \alpha_\theta - \frac{L_\theta \alpha_\theta}{\sqrt{K}}\right\}$, and $\kappa$ is a constant independent of $K$, which is defined in Eq. (73).*

*Proof.* The proof is provided in Appendix A.1, which adapts stochastic gradient ascent (SGA) analysis to our coupled policy-dynamics update, accounting for the interdependence between the policy parameters $\theta$ and the ODE dynamic parameters $\phi$. □

Theorem 4.2 establishes that the proposed joint training algorithm for the policy and the neural ODE dynamics converges to a stationary point of the objective $J(\theta, \phi)$ at a rate of $\tilde{\mathcal{O}}(1/\sqrt{K})$ in expectation, where $K$ is the total number of optimization iterations. Under the chosen learning rates $\eta_\theta = \alpha_\theta/\sqrt{K}$ and $\eta_\phi = \alpha_\phi/\sqrt{K}$, the algorithm provably finds an $\epsilon$-stationary point within $\mathcal{O}(1/\epsilon^2)$ iterations, which matches the rate of standard stochastic gradient (SGD) methods in non-convex optimization despite the additional complexity of learning the continuous-time latent dynamics.

### 4.3. Policy Optimization with KL Regularization

To preserve the capabilities of discrete-time language models and maintain training stability, we formulate a regularization term using Kullback-Leibler divergence between the learning policy $\pi$ and a pre-trained language model (PLM), e.g., LLaMA (Touvron et al., 2023) and Qwen (Bai et al.,

2023). The optimization objective can be formulated by:

$$
\begin{aligned}
J_{\mathrm{KL}}(\pi) =& \mathbb{E}_{x \sim \mu}\Bigg[ \mathbb{E}_{\pi}\Big[ \sum_{t=0}^{T-1} \gamma^t r(X_t^{\pi}, a_t^{\pi}) \mid X_0^x \Big] \\
& - \beta \mathrm{KL}\big( \mathbb{P}(\cdot \mid x, \pi) \parallel \mathbb{P}(\cdot \mid x, \pi_{\mathrm{ref}}) \big) \Bigg],
\end{aligned}
\tag{10}
$$

where $\mathrm{KL}(\cdot \parallel \cdot)$ denotes the KL divergence, such that for two distributions $p_1$ and $p_2$, KL divergence is formulated as $\mathrm{KL}(p_1 \parallel p_2) = \mathbb{E}_{x \sim p_1}[\log(p_1(x)/p_2(x))]$. $\mathbb{P}(\cdot \mid x, \pi)$ denotes the distribution over action sequences $\{a_t^{\pi}\}_{t=0}^{T-1}$ induced by the policy $\pi$ starting from initial state $s_0 = X_0^x$ following the ODE dynamics in Eq. (1), and $\mathbb{P}(\cdot \mid x, \pi_{\mathrm{ref}})$ denotes the corresponding distribution under the reference policy, and $\beta > 0$ controls the strength of the regularization.

Although the ODE dynamics are continuous, tokens are emitted at discrete steps $t \in \{0, \ldots, T-1\}$. Before decomposing the following theorem shows that under the ODE dynamics, the sequence-level KL divergence decomposes into a sum of per-step divergences, leading to a tractable regularization loss.

**Definition 4.3** (**Mapping Latent States to Token Sequences**). We define a mapping $h : \mathbb{R}^d \to \bigcup_{t=0}^{T-1} \mathcal{A}^t$ where $\mathcal{A}^t = \{(x, a_0^{\pi}, \ldots, a_{t-1}^{\pi}) : x \in \mathcal{X}, a_i^{\pi} \in \mathcal{A}\}$ for $t \geq 1$, and $\mathcal{A}^0 = \{(x, \emptyset) : x \in \mathcal{X}\}$ (the input only, with no generated tokens). The mapping $h$ associates each latent state $X_t^{\pi}$ with the prompt and the sequence of input and previously generated tokens $(x, a_{<t}^{\pi}) = h(X_t^{\pi})$.

Consequently, the reference policy $\pi_{\mathrm{ref}}$ can be expressed as a function of the latent state via this definition of mapping. We have the following theorem:

**Theorem 4.4** (**KL Decomposition**). *Under the ODE dynamics in Eq. (1) and Definition 4.3, the KL divergence between the action sequence distributions induced by policy $\pi$ and reference policy $\pi_{\mathrm{ref}}$ satisfies:*

$$
\begin{aligned}
& \mathrm{KL}\left( \mathbb{P}(\cdot \mid x, \pi) \parallel \mathbb{P}(\cdot \mid x, \pi_{\mathrm{ref}}) \right) \\
=& \mathbb{E}_{\pi}\left[ \sum_{t=0}^{T-1} \mathrm{KL}\left( \pi(\cdot \mid X_t^{\pi}) \parallel \pi_{\mathrm{ref}}(\cdot \mid h(X_t^{\pi})) \right) \Big| X_0^x \right],
\end{aligned}
$$

*where the expectation is taken over trajectories generated by the policy $\pi$ and the ODE dynamics.*

*Proof.* The full proof is provided in Appendix A.2. □

Theorem 4.4 allows us to simplify the original objective in Eq. (10). By treating the sequence-level KL divergence constraint as a step-level regularization term. Accordingly, we can define the corresponding value function for our RL problem as:

$$
\begin{aligned}
V^{\pi}(\mu) = & \mathbb{E}_{x \sim \mu}\mathbb{E}_{\pi}\Bigg[ \sum_{t=0}^{T-1} \Big( \gamma^t r(X_t^{\pi}, a_t^{\pi}) \\
& - \beta \mathrm{KL}\left( \pi(\cdot \mid X_t^{\pi}) \parallel \pi_{\mathrm{ref}}(\cdot \mid h(X_t^{\pi})) \right) \Big) \Big| X_0^x \Bigg],
\end{aligned}
\tag{11}
$$

where $\beta > 0$ is the regularization coefficient from Eq. (10).

The following theorem establishes a suboptimality bound that quantifies the performance gap between the optimal value and the value achieved in practice.

**Theorem 4.5** (**Suboptimality Bound**). *Let $\pi^* = \arg\max_{\pi} V^{\pi}(\mu)$ denote the optimal policy and $\hat{\pi} = \arg\max_{\pi} J_{\mathrm{KL}}(\pi)$ denote the KL-regularized optimal policy. Under the assumption that the reward is bounded, i.e., $r \in [0, 1]$, the suboptimality bound for any policy $\pi$ can be expressed as:*

$$
\begin{aligned}
& V^{\pi^*}(\mu) - V^{\pi}(\mu) \\
\leq & \sum_{t=0}^{T-1}\Bigg( \beta \mathbb{E}_{x \sim \mu}\mathbb{E}_{\pi^*}\Big[ \mathrm{KL}\big( \pi^*(\cdot \mid X_t^{\pi^*}) \parallel \pi_{\mathrm{ref}}(\cdot \mid h(X_t^{\pi^*})) \big) \Big| X_0^x \Big] \\
& + \frac{1}{1-\gamma}\sqrt{2 \mathbb{E}_{x \sim \mu}\mathbb{E}_{\hat{\pi}}\left[ \mathrm{KL}\left( \hat{\pi}(\cdot \mid X_t^{\hat{\pi}}) \parallel \pi(\cdot \mid X_t^{\hat{\pi}}) \right) \mid X_0^x \right]} \Bigg).
\end{aligned}
$$

*Proof.* The proof is provided in Appendix A.3. □

This theorem characterizes the performance gap between the true optimal policy $\pi^*$ and any candidate policy $\pi$. The bound consists of two key components: The first term reflects the *regularization bias*, determined by the divergence of $\pi^*$ from the reference policy $\pi_{\mathrm{ref}}$, scaled by $\beta$. This indicates that greater deviation from the reference policy can inherently limit the maximum achievable value. The second term represents the *approximation error*, which depends on the KL divergence between the learned policy $\pi$ and the KL-regularized optimal policy $\hat{\pi}$, suggesting that keeping $\pi$ close to $\hat{\pi}$ is sufficient to guarantee near-optimal performance. This theoretical insight motivates our algorithmic design, which explicitly minimizes this divergence during training.

### 4.4. Neural Architecture Design

We introduce two core neural components that enable efficient and scalable learning of continuous latent dynamics for text generation: a scaled neural ODE dynamics network and a hierarchical policy network.

**Scaled neural ODE dynamics network.** We design a time-conditioned residual network to parameterize the dynamics $b_{\phi}$ in the neural ODE (Eq. (4)). The network takes the latent state $X_s^{\theta} \in \mathbb{R}^d$, a sinusoidal time encoding $\mathcal{E}_t(s) \in \mathbb{R}^d$, and an action embedding $\mathcal{E}_a(a_t^{\theta}) \in \mathbb{R}^d$ as input, producing

*Table 1.* Memory and computational complexity in comparison with the standard Transformer.

| Complexity | Latent ODE | Transformer |
|---|---|---|
| Memory | $\mathcal{O}(d)$ | $\mathcal{O}(T \cdot d)$ |
| Computation | $\mathcal{O}(T \cdot N \cdot d^2)$ | $\mathcal{O}(T^2 \cdot d + T \cdot d^2)$ |

the infinitesimal change $dX_s^\theta/ds$. The whole architecture is a multi-layer feed forward network (FFN). To ensure sufficient representation capacity for modeling complex text dynamics, we scale the architecture with two key design choices: (1) expanded hidden dimensions, and (2) deep residual stacks. The architecture is defined as:

$$h^{(0)} = \text{LayerNorm}\left(X_s + \mathcal{E}_t(s) + \mathcal{E}_a(a_t^\theta)\right)$$

For $l = 1, \ldots, L_n$ :

$$h^{(l)} = h^{(l-1)} + \text{FFN}_{\text{N-ODE}}^{(l)}\left(\text{RMSNorm}(h^{(l-1)}), \mathcal{E}_t(s)\right)$$

$$b_\phi(s, X_s^\theta, a_t^\theta) = \text{Linear}_{\mathbb{R}^d \to \mathbb{R}^d}\left(h^{(L_n)}\right),$$

where each feed forward layer $\text{FFN}_{\text{N-ODE}}^{(l)}$ employs expansion with ratio $r_n$: $\text{FFN}_{\text{N-ODE}}^{(l)}(x, y) = \text{Linear}_{\mathbb{R}^{d \cdot r_n} \to \mathbb{R}^d}(\text{SwiGLU}(\text{Linear}_{\mathbb{R}^d \to \mathbb{R}^{d \cdot r_n}}(x + y)))$ with SwiGLU activation $\text{SwiGLU}(y) = \text{Swish}(y_{:d \cdot r_n/2}) \odot y_{d \cdot r_n/2:}$.

**Hierarchical policy network.** The policy network $\pi_\theta(\cdot \mid X_t^\theta)$ is implemented as a deep residual network with $L_p$ blocks to better map latent states to action distributions:

$$h^{(0)} = X_t^\theta$$

For $l = 1, \ldots, L_p$ :

$$h^{(l)} = h^{(l-1)} + \text{FFN}_{\text{policy}}^{(k)}\left(\text{RMSNorm}(h^{(l-1)})\right)$$

$$\pi_\theta(a \mid X_t^\theta) = \text{softmax}\left(\frac{W \cdot h^{(L_p)}}{\tau}\right),$$

where each feed forward layer $\text{FFN}_{\text{policy}}^{(k)}$ employs expansion with ratio $r_p$: $\text{FFN}_{\text{policy}}^{(k)}(x) = \text{Linear}_{\mathbb{R}^{d \cdot r_p} \to \mathbb{R}^d}(\text{GeGLU}(\text{Linear}_{\mathbb{R}^d \to \mathbb{R}^{d \cdot r_p}}(x)))$ with GeGLU activation $\text{GeGLU}(x) = \text{GELU}(x_{:d \cdot r_p/2}) \odot x_{d \cdot r_p/2:}$, $W \in \mathbb{R}^{|\mathcal{V}| \times d}$ is the output projection matrix where $|\mathcal{V}|$ denotes the vocabulary size and $\tau > 0$ is a learnable temperature parameter.

**Memory and computational efficiency.** Our framework introduces a memory and computation efficient architecture. We compare with the standard Transformer structure (Vaswani et al., 2017) based on the following notations: $T$ = sequence length, $d$ = latent dimension and $N$ = number of ODE steps per token interval. For memory efficiency, the ODE solver, e.g., in Eq. (6) required a memory cost of $\mathcal{O}(d)$ to store the latent state of the previous step. This cost is independent of the number of solver step size $N$ and sequence length $T$, and significantly smaller than standard Transformers of $\mathcal{O}(T \cdot d)$, where the states of the whole sequence should be stored for the self-attention computation. The computational efficiency of our latent ODE structure for one sample in the forward pass scales as $\mathcal{O}(T \cdot N \cdot d^2)$. Crucially, we avoid the $\mathcal{O}(T^2)$ self-attention cost in the standard Transformer structure, making our method efficient for long context.

## 5. Experiments

We evaluate our proposed continuous-time latent dynamics method on standard text generation tasks, comparing against discrete-time Transformer baselines across policy optimization strategies and generation quality metrics.

### 5.1. Experimental Setup

**Dataset.** We use the *UltraFeedback-binarized* [1] dataset, a pre-processed version of the original UltraFeedback dataset (Cui et al., 2024). This dataset is a widely used benchmark for RLHF and contains approximately 64,000 prompts sourced from diverse domains such as question answering, summarization, and dialogue generation. Each instance consists of a context (prompt), two model responses, and a binary preference label that indicates the better response. We evaluate all methods on a widely adopted LLM-as-a-Judge benchmark: AlpacaEval 2 (Li et al., 2023), which uses general human instructions from AlpacaFarm (Dubois et al., 2023) and compares model responses against GPT-4-Turbo references via a GPT-4-Turbo judge, reporting ROUGE-L, raw win rate (WR) and the length-controlled win rate (LC) (Dubois et al., 2024) over the reference responses.

**Neural network construction.** We design five CT-LLM variants spanning from 0.4B to 2.1B parameters to study scaling properties. Table 2 provides detailed specifications. The main CT-LLM-Large (1.2B) configuration uses latent dimension $d = 2048$, $L_n = 12$ ODE blocks with expansion ratio $r_n = 4$, and $L_p = 6$ policy blocks with $r_p = 4$. We set the temperature parameter $\tau = 1.0$ for the softmax function in hierarchical policy network. The reference policy $\pi_{\text{ref}}$ is a frozen large language model `Llama-3-8B-Instruct`[2].

We make comparison with the standard discrete-time Transformer baseline (Vaswani et al., 2017) in our experiments. To ensure a fair comparison, we meticulously align the key hyperparameters between CT-LLM and the Transformer baselines (e.g., vocabulary size, layer normalization, activa-

---

[1] https://huggingface.co/datasets/HuggingFaceH4/ultrafeedback_binarized.

[2] https://huggingface.co/meta-llama/Meta-Llama-3-8B-Instruct.

*Table 2.* Architecture specifications for CT-LLM variants. Notations: $d$ = latent dimension, $L_n$ = number of ODE blocks, $r_n$ = ODE expansion ratio, $L_p$ = number of policy blocks, $r_p$ = policy expansion ratio.

| Variant | Params | $d$ | $L_n$ | $r_n$ | $L_p$ | $r_p$ |
|---|---|---|---|---|---|---|
| CT-LLM-Small | 0.4B | 1024 | 8 | 4 | 4 | 4 |
| CT-LLM-Medium | 0.7B | 1536 | 10 | 4 | 5 | 4 |
| CT-LLM-Large | 1.2B | 2048 | 12 | 4 | 6 | 4 |
| CT-LLM-XLarge | 2.1B | 2048 | 16 | 8 | 8 | 4 |

*Table 3.* Results of AlpacaEval 2 across different metrics. We denote ROUGE-L (R-L), raw win rate (WR) and length-controlled win rate (LC) in percentage (%) respectively. We format **the best**, the second best results.

| Arch. | Strategy | Params | R-L | WR | LC |
|---|---|---|---|---|---|
| Llama-3-8B-Instruct (*Ref.*) | | 8.0B | $40.7_{\pm0.3}$ | $22.7_{\pm0.3}$ | $23.2_{\pm0.4}$ |
| Transformer (Dec-only) | RL | 1.7B | $29.5_{\pm0.4}$ | $11.5_{\pm0.6}$ | $13.1_{\pm0.7}$ |
| | RL+KL | 1.7B | $\underline{32.5_{\pm0.4}}$ | $11.4_{\pm0.7}$ | $12.8_{\pm0.5}$ |
| Transformer (Enc-Dec) | RL | 1.7B | $25.6_{\pm0.2}$ | $5.7_{\pm0.4}$ | $6.2_{\pm0.3}$ |
| | RL+KL | 1.7B | $27.2_{\pm0.3}$ | $7.4_{\pm0.4}$ | $8.9_{\pm0.5}$ |
| ODE-Policy | **CT-LLM** (RL+KL) | 1.2B | $\mathbf{36.5_{\pm0.7}}$ | $\mathbf{16.6_{\pm0.3}}$ | $\mathbf{19.2_{\pm0.4}}$ |
| | w/o KL | 1.2B | $30.5_{\pm0.8}$ | $\underline{12.6_{\pm0.4}}$ | $\underline{15.4_{\pm0.3}}$ |

*Table 4.* Performance and efficiency scaling for CT-LLM variants on AlpacaEval 2.

| Variant | Params | R-L $\uparrow$ | LC $\uparrow$ | PPL $\downarrow$ |
|---|---|---|---|---|
| CT-LLM-Small | 0.4B | $18.7_{\pm1.1}$ | $3.6_{\pm0.3}$ | 9.8 |
| CT-LLM-Medium | 0.7B | $30.3_{\pm0.6}$ | $10.5_{\pm0.4}$ | 8.1 |
| CT-LLM-Large (Main) | 1.2B | $36.8_{\pm0.7}$ | $19.2_{\pm0.4}$ | 7.2 |
| CT-LLM-XLarge | 2.1B | $38.1_{\pm0.5}$ | $22.0_{\pm0.7}$ | 6.8 |
| Llama-3-8B-Instruct (*Ref.*) | 8.0B | $40.7_{\pm0.3}$ | $23.2_{\pm0.4}$ | 5.9 |

Transformer architecture following Llama (Touvron et al., 2023) and Qwen (Bai et al., 2023), and (ii) Transformer (Enc-Dec): an encoder-decoder Transformer architecture (Vaswani et al., 2017). Both baseline models are first pre-trained on the same dataset via standard supervised learning. For each baseline architecture, we use different optimization strategies: RL: Uses a policy optimization algorithm based on the policy gradient method of Eq. (8); RL+KL: Employs a KL divergence regularization strategy as Eq. (10) for training stability.

**Main results.** As illustrated in Table 3, our proposed method, CT-LLM, consistently outperforms the discrete-time architecture baselines on all evaluation metrics. This demonstrates the effectiveness of modeling text generation as a continuous-time latent dynamical process optimized via reinforcement learning. Results show that our continuous-time formulation consistently outperforms the discrete architectures across different training strategies (Pure RL (w/o KL) and Joint RL+KL) on the AlpacaEval 2 benchmark. Specifically, CT-LLM achieves the best ROUGE-L scores, which is a notable over 4% absolute improvement over the best-performing baselines. More importantly, on the more indicative WR and LC metrics, our method outperforms the strongest baselines by approximately 3% to 4%. This demonstrates the superiority of our continuous-time framework across different metrics, particularly in the joint RL+KL setting where the latent dynamics can better leverage the KL regularization for training stability. Furthermore, the performance gap between our method and the baselines is consistent across both the decoder-only and encoder-decoder architectures Transformer models. This suggests that the benefits of the continuous latent dynamics formulation are architecture-agnostic and stem from the fundamental modeling advantage of smoother state transitions and stable policy optimization.

**Performance and scaling analysis.** As shown in Table 4, our CT-LLM variants demonstrate consistent performance scaling with increasing model parameters on the AlpacaEval 2 benchmark. Specifically, the length-controlled win rate (LC) improves from 3.6% (0.4B) to 22.0% (2.1B), and the perplexity (PPL) decreases correspondingly, indicating enhanced generation quality and fluency. The gains exhibit a pattern of diminishing returns beyond the 1.2B-parameter

tion function, etc), detailed configurations are illustrated in Appendix B.

**Optimization protocol.** We employ a two-phase optimization strategy. In the first *supervised pre-training phase*, we fix the ODE parameters $\phi$ and train only the policy $\pi_\theta$ on the instruction dataset using maximum likelihood estimation, minimizing the cross-entropy loss with respect to the reference model's outputs. This phase ensures the policy network produces reasonable token distributions before RL fine-tuning. The second *reinforcement learning phase* jointly optimizes both $\theta$ and $\phi$ using the objective in Eq. (5). We use the policy gradient algorithm for policy updates, and the continuous adjoint sensitivity method (Chen et al., 2018) to compute gradients through the ODE solver.

The reward function is a sentence-level reward from a pre-trained reward model (Christiano et al., 2017; Ouyang et al., 2022). The coefficient $\beta$ for KL-divergence regularization, as illustrated in Eq. (10), is annealed linearly from 0.1 to 0.01 during training. All models are trained with the AdamW optimizer using a linear warmup of 10% of the total steps followed by cosine decay. Gradient norms are clipped to 1.0 for stability. Appendix B illustrates the detailed training hyperparameters.

### 5.2. Results

We conduct experiments to evaluate the proposed continuous-time latent dynamical model for text generation. For comparisons, we consider two predominant Transformer architectures: (i) Transformer (Dec-only): a decoder-only

*Table 5.* Compute-matched comparison with the standard Transformer baselines.

| Model | Params | Training FLOPs | Training Iterations | ROUGE-L |
|---|---|---|---|---|
| CT-LLM (Ours) | 1.2B | $\approx 1.4 \times 10^{20}$ | 50,000 | **36.8** |
| Transformer | 1.7B | $\approx 1.4 \times 10^{20}$ | 40,000 | 28.5 |
| Transformer | 8B | $\approx 1.4 \times 10^{20}$ | 6,000 | 30.2 |

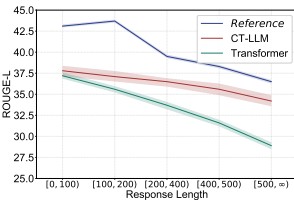

*(a)* ROUGE-L vs resp. length.

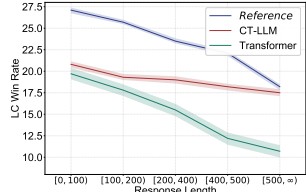

*(b)* LC vs Response Length.

*Figure 3.* Performance metrics as a function of response length.

scale (CT-LLM-Large), where performance begins to approach that of the much larger 8.0B-parameter reference model.

**Compute efficiency.** We fixed the total computational budget to approximately $\approx 1.4 \times 10^{20}$ FLOPs. Under this constraint, we consider a 1.7B Transformer baseline trained for 40,000 iterations and a 8B Transformer baseline trained for 6,000 iterations. As shown in Table 5, CT-LLM achieves superior performance the same compute constraints. This demonstrates the significantly higher compute efficiency of CT-LLM. Even when competing against a larger 8B Transformer model under the same FLOPs budget (where the 8B model is forced to underfit due to limited data exposure), our method achieves a substantial gain of 6.6% R-L score on AlpacaEval. This confirms that our architecture learns more effectively per unit of computation.

**Long-context generation.** To evaluate the capability of our method in generating long and coherent responses, we analyze model performance across different response length intervals, as shown in Figure 3. As the response length increases, all methods experience a performance decline, as measured by both ROUGE-L and length-controlled (LC) win rate. However, CT-LLM shows the least degradation, especially in the most challenging long-context regime ($[500, \infty)$ tokens). It retains an LC win rate of 17.5%, significantly higher than discrete Transformer baselines (10.7%) and close to the 8B reference model (18.2%). This indicates that the continuous-time latent dynamics provide a more stable state evolution over long sequences, mitigating the common issue observed in discrete-time auto-regressive models when generating long text. The smaller performance gap between CT-LLM and the reference model in long-context settings further suggests that our approach effectively preserves the reference model's long-range modeling capability with a KL regularization strategy.

# 6. Conclusion

We investigated a novel framework that models text generation as continuous-time latent dynamics optimized via reinforcement learning. Specifically, we conceptualize the process as a discrete-time Markov Decision Process (MDP) whose state evolves continuously in a low-dimensional latent space governed by a neural ODE. Our method co-optimizes this dynamical system and a token-emission policy to maximize a composite objective, demonstrating both provable theoretical guarantee and strong empirical performance, and offering a new paradigm for fluid and coherent sequence generation.

## Acknowledgments

We thank the anonymous reviewers for their helpful comments and suggestions.

## Impact Statement

This paper presents work whose goal is to advance the field of machine learning for natural language generation. We propose a novel framework that models text generation as continuous-time latent dynamics, aiming to improve the coherence and efficiency of language models. As with any generative technology, there exists a potential for misuse, such as in the creation of misleading content. We encourage the adoption of responsible AI practices to mitigate such risks.

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

# A. Proofs

## A.1. Proof of Theorem 4.2

**Assumption A.1 (Boundness).** We suppose that $b_\phi(s, X, a_t)$, $\frac{\partial b_\phi}{\partial \phi}$ and $\frac{\partial J}{\partial X_t}$ are boundness for any $t \in \{0, \ldots, T-1\}$. Specifically,

- There exists $B \geq 0$, such that $\|b_\phi(s, X, a_t)\| \leq B$.

- There exists $B_{\partial \phi} \geq 0$, such that $\|\frac{\partial b_\phi}{\partial \phi}\| \leq B_{\partial \phi}$.

- There exists $B_{\partial X} \geq 0$, such that $\|\frac{\partial J}{\partial X_t}\| \leq B_{\partial X}$.

**Assumption A.2 (Lipschitzness).** We suppose that neural ODE function $b_\phi(s, X, a_t)$ is Lipschitz continuous for any $t \in \{0, \ldots, T-1\}$. Specifically,

- $b_\phi(s, X, a_t)$ is $M_t$-Lipschitzness w.r.t. $s$, i.e., for any $s_1, s_2 \in (t, t+1]$, $\|b_\phi(s_1, X, a_t) - b_\phi(s_2, X, a_t)\| \leq M_t |s_1 - s_2|$.

- $b_\phi(s, X, a_t)$ is $L_b$-Lipschitzness w.r.t. $X$, i.e., for any $X_1, X_2$, $\|b_\phi(s, X_1, a_t) - b_\phi(s, X_2, a_t)\| \leq L_b \|X_1 - X_2\|$.

We suppose that $\frac{\partial b_\phi}{\partial \phi}(s, X, a_t)$ is Lipschitz continuous for any $t \in \{0, \ldots, T-1\}$. Specifically,

- $\frac{\partial b_\phi}{\partial \phi}(s, X, a_t)$ is $L_t$-Lipschitzness w.r.t. $s$, i.e., for any $s_1, s_2 \in (t, t+1]$,

$$\left| \frac{\partial b_\phi}{\partial \phi}(s_1, X, a_t) - \frac{\partial b_\phi}{\partial \phi}(s_2, X, a_t) \right| \leq L_t \|s_1 - s_2\|.$$

- $\frac{\partial b_\phi}{\partial \phi}(s, X, a_t)$ is $L_{\partial \phi}$-Lipschitzness w.r.t. $X$, i.e., for any $X_1, X_2$,

$$\left| \frac{\partial b_\phi}{\partial \phi}(t_n, X_1, a_t) - \frac{\partial b_\phi}{\partial \phi}(t_n, X_2, a_t) \right| \leq L_{\partial \phi} \|X_1 - X_2\|,$$

We suppose that $\frac{\partial b_\phi}{\partial X}(s, X, a_t)$ is Lipschitz continuous for any $t \in \{0, \ldots, T-1\}$. Specifically,

- $\frac{\partial b_\phi}{\partial X}(s, X, a_t)$ is $L_t$-Lipschitzness w.r.t. $s$, i.e., for any $s_1, s_2 \in (t, t+1]$,

$$\left| \frac{\partial b_\phi}{\partial X}(s_1, X, a_t) - \frac{\partial b_\phi}{\partial X}(s_2, X, a_t) \right| \leq L_t' \|s_1 - s_2\|.$$

- $\frac{\partial b_\phi}{\partial X}(s, X, a_t)$ is $L_{\partial \phi}$-Lipschitzness w.r.t. $X$, i.e., for any $X_1, X_2$,

$$\left| \frac{\partial b_\phi}{\partial X}(t_n, X_1, a_t) - \frac{\partial b_\phi}{\partial X}(t_n, X_2, a_t) \right| \leq L_{\partial X} \|X_1 - X_2\|,$$

We suppose that $\frac{\partial J(\theta, \phi)}{\partial \phi}$ is $L_J$-Lipschitzness w.r.t. $\theta$, i.e., for any $\theta_1, \theta_2$,

$$\left\| \frac{\partial J(\theta_1, \phi)}{\partial \phi} - \frac{\partial J(\theta_2, \phi)}{\partial \phi} \right\| \leq L_J \|\theta_1 - \theta_2\|.$$

**Assumption A.3 (Smoothness of $J(\theta, \phi)$).** We suppose that the objective of $J(\theta, \phi)$ in Eq. (5) is smoothness. Specifically,

- $J(\theta, \phi)$ is $L_\theta$-smoothness w.r.t. $\theta$.

- $J(\theta, \phi)$ is $L_\phi$-smoothness w.r.t. $\phi$.

**Assumption A.4 (Bounded Variance of $\nabla_\theta J(\theta, \phi)$).** We suppose the variance is bounded, such that for any $\theta$ and $\phi$,
$\mathbb{E}\left[\|\hat{\nabla}_\theta J(\theta, \phi) - \nabla_\theta J(\theta, \phi)\|^2\right] \leq \sigma^2$.

**Assumption A.5 (Bounded Variance of $\frac{\partial J(\theta, \phi)}{\partial \phi}$).** We suppose the variance is bounded, such that for any $\theta$ and $\phi$,
$\mathbb{E}\left[\left\|\frac{\partial J(\theta, \phi)}{\partial \phi} - \frac{\partial \hat{J}(\theta, \phi)}{\partial \phi}\right\|^2\right] \leq (\sigma')^2$.

**Lemma A.6 (Convergence of Policy Gradient Method).** *Under Assumptions A.3 & A.4, and based on the updating rule $\theta_{k+1} = \theta_k + \eta_\theta \hat{\nabla}_\theta J(\theta_k, \phi)$ for any $\phi$ at the $k$-th iteration, then we have*

$$\left(\eta_\theta - L_\theta \eta_\theta^2\right) \|\nabla_\theta J(\theta_k, \phi)\|^2 \leq \mathbb{E}\left[J(\theta_{k+1}, \phi) - J(\theta_k, \phi) \mid \mathcal{F}_k\right] + L_\theta \eta_\theta^2 \sigma^2, \tag{12}$$

*where $\{\mathcal{F}_k\}_{k \geq 0}$ denotes a filtration such that $\mathcal{F}_k$ denote the $\sigma$-algebra encoding all information available up to the $k$-th iteration.*

*Proof.* We denote the true policy gradient $\nabla_\theta J(\theta_k, \phi)$ and the empirical gradient $\hat{\nabla}_\theta J(\theta_k, \phi)$ from the update rule of SGA for any $\phi$ at the $k$-th training iteration is represented as Eq. (8) and Eq. (9), respectively.

Using the $L_\theta$-smoothness assumption of $J(\theta, \phi)$ w.r.t. $\theta$ (Assumption A.3), we have

$$J(\theta_{k+1}, \phi) \geq J(\theta_k, \phi) + \langle \nabla_\theta J(\theta_k, \phi), \theta_{k+1} - \theta_k \rangle - \frac{L_\theta}{2} \|\theta_{k+1} - \theta_k\|^2. \tag{13}$$

By the parameter updating rule: $\theta_{k+1} = \theta_k + \eta_\theta \hat{\nabla}_\theta J(\theta_k, \phi)$ for any $\phi$, we have

$$J(\theta_{k+1}, \phi) - J(\theta_k, \phi) \geq \eta_\theta \langle \nabla_\theta J(\theta_k, \phi), \hat{\nabla}_\theta J(\theta_k, \phi) \rangle - \frac{L_\theta}{2} \eta_\theta^2 \|\hat{\nabla}_\theta J(\theta_k, \phi)\|^2. \tag{14}$$

Rearranging the terms and taking the conditional expectation $\mathbb{E}\left[\cdot \mid \mathcal{F}_k\right]$ on both sides, where $\mathcal{F}_k$ denotes the $\sigma$-algebra encoding all information available up to the $k$-th iteration. Since $\hat{\nabla}_\theta J(\theta_k, \phi)$ is unbiased estimator of $\nabla_\theta J(\theta_k, \phi)$, it holds that

$$\eta_\theta \|\nabla_\theta J(\theta_k, \phi)\|^2 \leq \mathbb{E}\left[J(\theta_{k+1}, \phi) - J(\theta_k, \phi) \mid \mathcal{F}_k\right] + \frac{L_\theta}{2} \eta_\theta^2 \|\hat{\nabla}_\theta J(\theta_k, \phi)\|^2 \tag{15}$$

$$\leq \mathbb{E}\left[J(\theta_{k+1}, \phi) - J(\theta_k, \phi)\right] + L_\theta \eta_\theta^2 \left(\|\nabla_\theta J(\theta_k, \phi)\|^2 + \mathbb{E}\left[\|\hat{\nabla}_\theta J(\theta_k) - \nabla_\theta J(\theta_k, \phi)\|^2 \mid \mathcal{F}_k\right]\right) \tag{16}$$

$$\leq \mathbb{E}\left[J(\theta_{k+1}, \phi) - J(\theta_k, \phi)\right] + L_\theta \eta_\theta^2 \left(\|\nabla_\theta J(\theta_k, \phi)\|^2 + \sigma^2\right), \tag{17}$$

where the last inequality follows from Assumption A.4.

Rearranging the terms, we have

$$\left(\eta_\theta - L_\theta \eta_\theta^2\right) \|\nabla_\theta J(\theta_k, \phi)\|^2 \leq \mathbb{E}\left[J(\theta_{k+1}, \phi) - J(\theta_k, \phi) \mid \mathcal{F}_k\right] + L_\theta \eta_\theta^2 \sigma^2, \tag{18}$$

which completes the proof of Lemma A.6. $\square$

**Lemma A.7 (Convergence of Adjoint Gradient Method).** *Under Assumptions A.1 & A.2 & A.3 and assume that the number of steps satisfies $N \geq \frac{2\eta_\theta T \tilde{C}}{\left(\eta_\phi - \frac{3L_\phi \eta_\phi^2}{2}\right) \|\partial J(\theta, \phi)/\partial \phi\|}$, then for any $\theta$ at any iteration $k \in \{0, 1, \ldots, K-1\}$, we have*

$$\frac{1}{2}\left(\eta_\phi - \frac{3L_\phi \eta_\phi^2}{2}\right) \left\|\frac{\partial J(\theta, \phi_k)}{\partial \phi}\right\|^2 \leq \mathbb{E}\left[J(\theta, \phi_{k+1}) - J(\theta, \phi_k) \mid \mathcal{F}_k\right] + \frac{3L_\phi \eta_\phi^2 \left((\sigma')^2 + T^2 \tilde{C}^2 h^2\right)}{2}, \tag{19}$$

*where $\tilde{C}$ denotes a constant defined in Eq. (49) and $\{\mathcal{F}_k\}_{k \geq 0}$ denotes a filtration such that $\mathcal{F}_k$ denote the $\sigma$-algebra encoding all information available up to the $k$-th iteration.*

*Proof.* Following Hairer et al. (1993), we denote the error bound of an ODE solver using Euler method with the step size of $h$ between the true latent state $X_{t_n}$ and the estimated latent state $\hat{X}_{t_n}$ as

$$\|\hat{X}_{t_n} - X_{t_n}\| \le C_X h, \tag{20}$$

where the coefficient $C_X$ denotes a constant that will be defined later. Specifically, we bound the local error for nay $t \in \{0, 1, \ldots, T - 1\}$ as follows:

$$\text{Err}_{\text{local}} = \int_{t_n}^{t_{n+1}} (b_\phi(s, X_s, a_t) - b_\phi(t_n, X_{t_n}, a_t)) \, ds \le \int_{t_n}^{t_{n+1}} \|b_\phi(s, X_s, a_t) - b_\phi(t_n, X_{t_n}, a_t)\| \, ds \tag{21}$$

$$\le \int_{t_n}^{t_{n+1}} (\|b_\phi(s, X_s, a_t) - b_\phi(t_n, X_s, a_t)\| + \|b_\phi(t_n, X_s, a_t) - b_\phi(t_n, X_{t_n}, a_t)\|) \, ds. \tag{22}$$

Using Assumption A.2, we have $\|b_\phi(s, X_s, a_t) - b_\phi(t_n, X_s, a_t)\| \le M_t(s - t_n)$ and $\|b_\phi(t_n, X_s, a_t) - b_\phi(t_n, X_{t_n}, a_t)\| \le L_b \|X_s - X_{t_n}\|$.

By the Lagrange's mean value theorem and Assumption A.1, we have

$$\|X_{s_1} - X_{s_2}\| = \left\| \int_{s_1}^{s_2} b_\phi(s, X_s, a_t) ds \right\| \le \sup_{t_n < s \le t_{n+1}} \|b_\phi(s, X_s, a_t)\| |s_1 - s_2| \le B|s_1 - s_2|. \tag{23}$$

Then, we have

$$\|b_\phi(t_n, X_s, a_t) - b_\phi(t_n, X_{t_n}, a_t)\| \le BL_b(s - t_n). \tag{24}$$

Thus, the local error can be bounded as:

$$\text{Err}_{\text{local}} \le \int_{t_n}^{t_{n+1}} (M_t + BL_b)(s - t_n) ds = \frac{(M_t + BL_b)h^2}{2}. \tag{25}$$

Using the definition: $\hat{X}_{t_{n+1}} = \hat{X}_{t_n} + h \cdot b_\phi(t_n, \hat{X}_{t_n}, a_t)$ and $X_{t_{n+1}} = X_{t_n} + h \cdot b_\phi(t_n, X_{t_n}, a_t) + \text{Err}_{\text{local}}$, we have

$$\|\hat{X}_{t_{n+1}} - X_{t_{n+1}}\| \le \|\hat{X}_{t_n} - X_{t_n}\| + h\|b_\phi(t_n, \hat{X}_{t_n}, a_t) - b_\phi(t_n, X_{t_n}, a_t)\| + \frac{(M_t + BL_b)h^2}{2} \tag{26}$$

$$\le (1 + hL_b)\|\hat{X}_{t_n} - X_{t_n}\| + \frac{(M_t + BL_b)h^2}{2}. \tag{27}$$

We use recursion to obtain:

$$\|\hat{X}_{t_n} - X_{t_n}\| \le \frac{(M_t + BL_b)h^2}{2} \sum_{i=0}^{I-1} (1 + hL_b)^i \tag{28}$$

$$\le \frac{(M_t + BL_b)h^2}{2} \cdot \frac{(1 + hL_b)^I - 1}{hL_b} \le \frac{(M_t + BL_b)}{2L_b} (e^{hIL_b} - 1)h = \frac{(M_t + BL_b)}{2L_b} (e^{T'L_b} - 1)h =: C_X h, \tag{29}$$

where the second inequality follows from the fact that $(1 + x)^k \le e^{kx}$, $T'$ denotes the maximum total length of a combination of the prompt and response.

Similarly, we can bound the error for the adjont state $\hat{\lambda}_{t_n}$ as

$$\|\hat{\lambda}_{t_n} - \lambda_{t_n}\| \le C_\lambda h, \tag{30}$$

where $C_\lambda$ denotes a constant that will be defined later. Specifically, we define and bound the local error as follows:

$$\text{Err}_{\text{local}} = \int_{t_n}^{t_{n+1}} \left( \left( \frac{\partial b_\phi(t_{n+1}, X_{t_{n+1}}, a_t)}{\partial X_{t_{n+1}}} \right)^\top \lambda_{t_{n+1}} - \left( \frac{\partial b_\phi(s, X_s, a_t)}{\partial X_s} \right)^\top \lambda_s \right) ds$$

$$\le \int_{t_n}^{t_{n+1}} \left| \left( \frac{\partial b_\phi(t_{n+1}, X_{t_{n+1}}, a_t)}{\partial X_{t_{n+1}}} \right)^\top \lambda_{t_{n+1}} - \left( \frac{\partial b_\phi(s, X_s, a_t)}{\partial X_s} \right)^\top \lambda_s \right| ds$$

$$\le \int_{t_n}^{t_{n+1}} \left\| \frac{\partial b_\phi(t_{n+1}, X_{t_{n+1}}, a_t)}{\partial X_{t_{n+1}}} - \frac{\partial b_\phi(s, X_s, a_t)}{\partial X_s} \right\| \|\lambda_{t_{n+1}}\| ds + \int_{t_n}^{t_{n+1}} \left\| \frac{\partial b_\phi(s, X_s, a_t)}{\partial X_s} \right\| \|\lambda_{t_{n+1}} - \lambda_s\| ds.$$

By the triangle inequality, we have

$$\left\| \frac{\partial b_\phi(t_{n+1}, X_{t_{n+1}}, a_t)}{\partial X_{t_{n+1}}} - \frac{\partial b_\phi(s, X_s, a_t)}{\partial X_s} \right\|$$

$$\leq \left\| \frac{\partial b_\phi(t_n, X_{t_{n+1}}, a_t)}{\partial X_{t_{n+1}}} - \frac{\partial b_\phi(s, X_{t_{n+1}}, a_t)}{\partial X_{t_{n+1}}} \right\| + \left\| \frac{\partial b_\phi(s, X_{t_{n+1}}, a_t)}{\partial X_{t_{n+1}}} - \frac{\partial b_\phi(s, X_s, a_t)}{\partial X_s} \right\|$$

$$\leq L_t'(t_{n+1} - s) + L_{\partial X} \| X_{t_{n+1}} - X_s \| \leq (L_t' + B L_{\partial X})(t_{n+1} - s),$$

where the second inequality follows from Assumption A.2 the third inequality follows from Eq. (23).

Besides, we can bound $\|\lambda_s\|$ by proving the stability of adjoint equation of Eq. (7) (boundness of $\|\lambda_s\|$ for any $t \leq s \leq t+1$) as follows

$$\frac{\mathrm{d}}{\mathrm{d}s} \|\lambda_s\|^2 = \frac{\mathrm{d}}{\mathrm{d}s} (\lambda_s^\top \lambda_s) = 2\lambda_s^\top \frac{\mathrm{d}\lambda_s}{\mathrm{d}s} = -2\lambda_s^\top \left( \frac{\partial b_\phi}{\partial X_s}(s, X_s, a_t) \right)^\top \lambda_s = -2 \left( \frac{\partial b_\phi}{\partial X_s}(s, X_s, a_t)\lambda_s \right)^\top \lambda_s. \tag{31}$$

Then by Cauchy-Schwarz inequality and Assumption A.2, we have $\frac{\mathrm{d}}{\mathrm{d}s} \|\lambda_s\|^2 \leq 2L_b \|\lambda_s\|^2$.

By Grönwall's inequality with Assumption A.2, then for $t < s \leq t+1$, we have $\|\lambda_s\|^2 \leq \|\lambda_{t+1}\|^2 e^{2L_b(t+1-s)}$.

By Assumption A.1 and the fact that $t + 1 - s \leq 1$, we have

$$\|\lambda_s\| \leq \|\lambda_{t+1}\| e^{L_b} \leq B_{\partial X} e^{L_b}. \tag{32}$$

Thus, we have

$$\int_{t_n}^{t_{n+1}} \left\| \frac{\partial b_\phi(t_{n+1}, X_{t_{n+1}}, a_t)}{\partial X_{t_{n+1}}} - \frac{\partial b_\phi(s, X_s, a_t)}{\partial X_s} \right\| \|\lambda_{t_{n+1}}\| \mathrm{d}s \leq \int_{t_n}^{t_{n+1}} B_{\partial X} e^{L_b} (L_t' + B L_{\partial X})(t_{n+1} - s) \mathrm{d}s \tag{33}$$

$$= \frac{B_{\partial X} e^{L_b} (L_t' + B L_{\partial X})}{2} \cdot h^2. \tag{34}$$

By Jensen's inequality and Cauchy-Schwarz inequality, we have

$$\|\lambda_s - \lambda_{t_n}\| \leq \int_{t_n}^s \left\| \left( \frac{\partial b_\phi}{\partial X_{s'}}(s', X_{s'}, a_t) \right)^\top \lambda_{s'} \right\| \mathrm{d}s' \leq (s - t_n) \max_{t_n \leq s' \leq t_{n+1}} \left\| \frac{\partial b_\phi}{\partial X_{s'}}(s', X_{s'}, a_t) \right\| \cdot \|\lambda_{s'}\|$$

$$\leq (s - t_n) L_b B_{\partial X} e^{L_b}, \tag{35}$$

where the last step follows from Assumption A.2 and Eq. (32). This completes the derivation of Eq. (30).

Similarly, we have

$$\left\| \lambda_{t_{n+1}} - \lambda_s \right\| \leq (t_{n+1} - s) L_b B_{\partial X} e^{L_b}. \tag{36}$$

Then, by Assumption A.1, we have

$$\int_{t_n}^{t_{n+1}} \left\| \frac{\partial b_\phi(s, X_s, a_t)}{\partial X_s} \right\| \|\lambda_{t_{n+1}} - \lambda_s\| \mathrm{d}s \leq \int_{t_n}^{t_{n+1}} L_b B_{\partial X}^2 e^{L_b}(t_{n+1} - s) \mathrm{d}s = \frac{L_b B_{\partial X}^2 e^{L_b}}{2} \cdot h^2. \tag{37}$$

Thus, we have

$$\mathrm{Err}_{\mathrm{local}} \leq \frac{B_{\partial X} e^{L_b} (L_t' + B L_{\partial X} + L_b B_{\partial X})}{2} \cdot h^2. \tag{38}$$

Using the definition: $\hat{\lambda}_{t_n} = \hat{\lambda}_{t_{n+1}} + h \cdot \left( \frac{\partial b_\phi(t_{n+1}, \hat{X}_{t_{n+1}}, a_t)}{\partial \hat{X}_{t_{n+1}}} \right)^\top \hat{\lambda}_{t_{n+1}}$ and $\lambda_{t_n} = \lambda_{t_{n+1}} + h \cdot \left( \frac{\partial b_\phi(t_{n+1}, X_{t_{n+1}}, a_t)}{\partial X_{t_{n+1}}} \right)^\top \lambda_{t_{n+1}} +$ $\mathrm{Err}_{\mathrm{local}}$.

Then, we use the triangle inequality and Cauchy-Schwarz inequality,

$$
\begin{aligned}
\|\hat{\lambda}_{t_n} - \lambda_{t_n}\| \leq & \|\hat{\lambda}_{t_{n+1}} - \lambda_{t_{n+1}}\| + h \cdot \left| \left( \frac{\partial b_\phi(t_{n+1}, \hat{X}_{t_{n+1}}, a_t)}{\partial \hat{X}_{t_{n+1}}} \right)^\top \hat{\lambda}_{t_{n+1}} - \left( \frac{\partial b_\phi(t_{n+1}, X_{t_{n+1}}, a_t)}{\partial X_{t_{n+1}}} \right)^\top \lambda_{t_{n+1}} \right| + \mathrm{Err}_{\mathrm{local}} \\
\leq & \|\hat{\lambda}_{t_{n+1}} - \lambda_{t_{n+1}}\| + h \cdot \left| \left( \frac{\partial b_\phi(t_{n+1}, \hat{X}_{t_{n+1}}, a_t)}{\partial \hat{X}_{t_{n+1}}} \right)^\top (\hat{\lambda}_{t_{n+1}} - \lambda_{t_{n+1}}) \right| \\
& + h \cdot \left| \left( \frac{\partial b_\phi(t_{n+1}, \hat{X}_{t_{n+1}}, a_t)}{\partial \hat{X}_{t_{n+1}}} - \frac{\partial b_\phi(t_{n+1}, X_{t_{n+1}}, a_t)}{\partial X_{t_{n+1}}} \right)^\top \lambda_{t_{n+1}} \right| + \mathrm{Err}_{\mathrm{local}} \\
\leq & \|\hat{\lambda}_{t_{n+1}} - \lambda_{t_{n+1}}\| + h \cdot \left\| \frac{\partial b_\phi(t_{n+1}, \hat{X}_{t_{n+1}}, a_t)}{\partial \hat{X}_{t_{n+1}}} \right\| \cdot \|\hat{\lambda}_{t_{n+1}} - \lambda_{t_{n+1}}\| \\
& + h \cdot \left\| \frac{\partial b_\phi(t_{n+1}, \hat{X}_{t_{n+1}}, a_t)}{\partial \hat{X}_{t_{n+1}}} - \frac{\partial b_\phi(t_{n+1}, X_{t_{n+1}}, a_t)}{\partial X_{t_{n+1}}} \right\| \cdot \|\lambda_{t_{n+1}}\| + \mathrm{Err}_{\mathrm{local}} \\
\leq & (1 + hL_b)\|\hat{\lambda}_{t_{n+1}} - \lambda_{t_{n+1}}\| + L_{\partial X} B_{\partial X} e^{L_b} \|\hat{X}_{t_{n+1}} - X_{t_{n+1}}\| \cdot h + \mathrm{Err}_{\mathrm{local}} \\
\leq & (1 + hL_b)\|\hat{\lambda}_{t_{n+1}} - \lambda_{t_{n+1}}\| + \frac{B_{\partial X} e^{L_b}(2L_{\partial X} C_X + L_t' + BL_{\partial X} + L_b B_{\partial X})}{2} \cdot h^2,
\end{aligned}
$$

where the forth inequality follows from Assumption A.2 and Eq. (32), the last inequality follows from Eq. (20) and Eq. (38).

We use recursion to obtain:

$$
\|\hat{\lambda}_{t_n} - \lambda_{t_n}\| \leq \frac{B_{\partial X} e^{L_b}(2L_{\partial X} C_X + L_t' + BL_{\partial X} + L_b B_{\partial X}) \cdot h^2}{2} \sum_{i=0}^{I-1} (1 + hL_b)^i \tag{39}
$$

$$
\leq \frac{B_{\partial X} e^{L_b}(2L_{\partial X} C_X + L_t' + BL_{\partial X} + L_b B_{\partial X})}{2L_b}(e^{T'L_b} - 1)h =: C_\lambda h, \tag{40}
$$

where the second inequality follows from a similar derivation as in Eq. (29).

This completes the derivation of Eq. (36).

Then, we have:

$$
\begin{aligned}
& \left| \int_t^{t+1} \lambda_s^\top \frac{\partial b_\phi}{\partial \phi}(s, X_s, a_t)\mathrm{d}s - h \sum_{n=1}^N \hat{\lambda}_{t_n}^\top \frac{\partial b_\phi}{\partial \phi}(t_n, \hat{X}_{t_n}, a_t) \right| \\
\leq & \left| \int_t^{t+1} \lambda_s^\top \frac{\partial b_\phi}{\partial \phi}(s, X_s, a_t)\mathrm{d}s - h \sum_{n=1}^N \lambda_{t_n}^\top \frac{\partial b_\phi}{\partial \phi}(t_n, X_{t_n}, a_t) \right| + \left| h \sum_{n=1}^N \lambda_{t_n}^\top \frac{\partial b_\phi}{\partial \phi}(t_n, X_{t_n}, a_t) - h \sum_{n=1}^N \lambda_{t_n}^\top \frac{\partial b_\phi}{\partial \phi}(t_n, \hat{X}_{t_n}, a_t) \right| \\
& + \left| h \sum_{n=1}^N \left( \lambda_{t_n} - \hat{\lambda}_{t_n} \right)^\top \frac{\partial b_\phi}{\partial \phi}(t_n, \hat{X}_{t_n}, a_t) \right| =: (\mathrm{I}) + (\mathrm{II}) + (\mathrm{III}).
\end{aligned}
$$

**Step 1: Bounding** (I) . By Jensen's inequality and Cauchy-Schwarz inequality, we have

$$
\left| \int_t^{t+1} \lambda_s^\top \frac{\partial b_\phi}{\partial \phi}(s, X_s, a_t) \mathrm{d}s - h \sum_{n=1}^N \lambda_{t_n}^\top \frac{\partial b_\phi}{\partial \phi}(t_n, X_{t_n}, a_t) \right|
$$

$$
\leq \sum_{n=1}^N \left| \int_{t_n}^{t_{n+1}} \left( \lambda_s^\top \frac{\partial b_\phi}{\partial \phi}(s, X_s, a_t) - \lambda_{t_n}^\top \frac{\partial b_\phi}{\partial \phi}(t_n, X_{t_n}, a_t) \right) \mathrm{d}s \right|
$$

$$
\leq \sum_{n=1}^N \int_{t_n}^{t_{n+1}} \left| \lambda_s^\top \frac{\partial b_\phi}{\partial \phi}(s, X_s, a_t) - \lambda_{t_n}^\top \frac{\partial b_\phi}{\partial \phi}(t_n, X_{t_n}, a_t) \right| \mathrm{d}s
$$

$$
\leq \sum_{n=1}^N \int_{t_n}^{t_{n+1}} \left| \lambda_s^\top \frac{\partial b_\phi}{\partial \phi}(s, X_s, a_t) - \lambda_{t_n}^\top \frac{\partial b_\phi}{\partial \phi}(s, X_s, a_t) \right| + \left| \lambda_{t_n}^\top \frac{\partial b_\phi}{\partial \phi}(s, X_s, a_t) - \lambda_{t_n}^\top \frac{\partial b_\phi}{\partial \phi}(t_n, X_{t_n}, a_t) \right| \mathrm{d}s
$$

$$
\leq \sum_{n=1}^N \int_{t_n}^{t_{n+1}} \|\lambda_s - \lambda_{t_n}\| \cdot \left\| \frac{\partial b_\phi}{\partial \phi}(s, X_s, a_t) \right\| \mathrm{d}s + \sum_{n=1}^N \int_{t_n}^{t_{n+1}} \|\lambda_{t_n}\| \cdot \left\| \frac{\partial b_\phi}{\partial \phi}(s, X_s, a_t) - \frac{\partial b_\phi}{\partial \phi}(t_n, X_{t_n}, a_t) \right\| \mathrm{d}s.
$$

By Eq. (35), we have

$$
\|\lambda_s - \lambda_{t_n}\| \leq (s - t_n) L_b B_{\partial X} e^{L_b} \leq h L_b B_{\partial X} e^{L_b}. \tag{41}
$$

By Assumption A.1, we have

$$
\sum_{n=1}^N \int_{t_n}^{t_{n+1}} \|\lambda_s - \lambda_{t_n}\| \cdot \left\| \frac{\partial b_\phi}{\partial \phi}(s, X_s, a_t) \right\| \mathrm{d}s \leq h L_b B_{\partial X} e^{L_b} \cdot \max_{t_n \leq s \leq t_{n+1}} \left\| \frac{\partial b_\phi}{\partial \phi}(s, X_s, a_t) \right\| \leq h L_b B_{\partial X} B_{\partial \phi} e^{L_b}. \tag{42}
$$

Then, by the triangle inequality, we have

$$
\left\| \frac{\partial b_\phi}{\partial \phi}(s, X_s, a_t) - \frac{\partial b_\phi}{\partial \phi}(t_n, X_{t_n}, a_t) \right\| \leq \left\| \frac{\partial b_\phi}{\partial \phi}(s, X_s, a_t) - \frac{\partial b_\phi}{\partial \phi}(t_n, X_s, a_t) \right\| + \left\| \frac{\partial b_\phi}{\partial \phi}(t_n, X_s, a_t) - \frac{\partial b_\phi}{\partial \phi}(t_n, X_{t_n}, a_t) \right\|
$$

$$
\leq L_t |s - t_n| + L_{\partial \phi} \|X_s - X_{t_n}\| \leq (L_t + B L_{\partial \phi}) |s - t_n|,
$$

where the second inequality follows from Assumption A.2 and the third inequality follows from Eq. (23).

Then, by Eq. (32), we have

$$
\sum_{n=1}^N \int_{t_n}^{t_{n+1}} \|\lambda_{t_n}\| \cdot \left\| \frac{\partial b_\phi}{\partial \phi}(s, X_s, a_t) - \frac{\partial b_\phi}{\partial \phi}(t_n, X_{t_n}, a_t) \right\| \mathrm{d}s \leq \sum_{n=1}^N \int_{t_n}^{t_{n+1}} B_{\partial X} e^{L_b} (L_t + B L_{\partial \phi})(s - t_n) \mathrm{d}s
$$
$$
= \frac{h B_{\partial X} e^{L_b} (L_t + B L_{\partial \phi})}{2}. \tag{43}
$$

We can bound (I) by combining Eq. (42) & (43) as follows:

$$
\text{(I)} \leq h L_b B_{\partial X} B_{\partial \phi} e^{L_b} + \frac{h B_{\partial X} e^{L_b} (L_t + B L_{\partial \phi})}{2}. \tag{44}
$$

**Step 2: Bounding** (II). By Cauchy-Schwarz inequality, Jensen's inequality and Assumption A.2, we have

$$
\left| h \sum_{n=1}^N \lambda_{t_n}^\top \frac{\partial b_\phi}{\partial \phi}(t_n, X_{t_n}, a_t) - h \sum_{n=1}^N \lambda_{t_n}^\top \frac{\partial b_\phi}{\partial \phi}(t_n, \hat{X}_{t_n}, a_t) \right| \tag{45}
$$

$$
\leq h \|\lambda_{t_n}\| L_{\partial \phi} \sum_{n=1}^N \|X_{t_n} - \hat{X}_{t_n}\| \leq C_X B_{\partial X} e^{L_b} L_{\partial \phi} h, \tag{46}
$$

where the last step follows from Eq. (20) and Eq. (32).

**Step 3: bounding** (III). By Jensen's inequality and the Cauchy-Schwarz inequality, we have

$$\left| h \sum_{n=1}^{N} \left( \lambda_{t_n} - \hat{\lambda}_{t_n} \right)^\top \frac{\partial b_\phi}{\partial \phi}(t_n, \hat{X}_{t_n}, a_t) \right| \leq h \sum_{n=1}^{N} \left\| \frac{\partial b_\phi}{\partial \phi}(t_n, \hat{X}_{t_n}, a_t) \right\| \cdot \| \lambda_{t_n} - \hat{\lambda}_{t_n} \| \leq h B_{\partial \phi} C_\lambda, \tag{47}$$

where the last inequality follows from Assumption A.1 and Eq. (30).

Following from Step 1 to Step 3, we have

$$\left| \int_t^{t+1} \lambda_s^\top \frac{\partial b_\phi}{\partial \phi}(s, X_s, a_t) \mathrm{d}s - h \sum_{n=1}^{N} \hat{\lambda}_{t_n}^\top \frac{\partial b_\phi}{\partial \phi}(t_n, \hat{X}_{t_n}, a_t) \right| \tag{48}$$

$$\leq \left( L_b B_{\partial X} B_{\partial \phi} e^{L_b} + \frac{B_{\partial X} e^{L_b}(L_t + BL_{\partial \phi})}{2} + C_X B_{\partial X} e^{L_b} L_{\partial \phi} + B_{\partial \phi} C_\lambda \right) h =: \tilde{C} h. \tag{49}$$

Then, we have

$$\left\| \frac{\partial \hat{J}(\theta, \phi)}{\partial \phi} - \hat{\nabla}_\phi J(\theta, \phi) \right\| \leq \sum_{t=0}^{T-1} \left| \int_t^{t+1} \lambda_s^\top \frac{\partial b_\phi}{\partial \phi}(s, X_s, a_t) \mathrm{d}s - h \sum_{n=1}^{N} \hat{\lambda}_{t_n}^\top \frac{\partial b_\phi}{\partial \phi}(t_n, \hat{X}_{t_n}, a_t) \right| \leq T \tilde{C} h. \tag{50}$$

By the $L_\phi$-smoothness of $J(\theta, \phi)$ w.r.t. $\phi$ (Assumption A.3), we have

$$J(\theta, \phi_{k+1}) \geq J(\theta, \phi_k) + \langle \frac{\partial J(\theta, \phi_k)}{\partial \phi}, \phi_{k+1} - \phi_k \rangle - \frac{L_\phi}{2} \| \phi_{k+1} - \phi_k \|^2. \tag{51}$$

Then by the updating rule $\phi_{k+1} = \phi_k + \eta_\phi \hat{\nabla}_\phi J(\theta, \phi_k)$ for any $\theta$, we have

$$J(\theta, \phi_{k+1}) \geq J(\theta, \phi_k) + \eta_\phi \langle \frac{\partial J(\theta, \phi_k)}{\partial \phi}, \hat{\nabla}_\phi J(\theta, \phi_k) \rangle - \frac{L_\phi}{2} \eta_\phi^2 \| \hat{\nabla}_\phi J(\theta, \phi_k) \|^2. \tag{52}$$

On the other hand, using the Cauchy-Schwarz inequality, we have

$$\langle \frac{\partial J(\theta, \phi_k)}{\partial \phi}, \hat{\nabla}_\phi J(\theta, \phi_k) \rangle = \langle \frac{\partial J(\theta, \phi_k)}{\partial \phi}, \frac{\partial \hat{J}(\theta, \phi_k)}{\partial \phi} \rangle - \langle \frac{\partial J(\theta, \phi_k)}{\partial \phi}, \frac{\partial \hat{J}(\theta, \phi_k)}{\partial \phi} - \hat{\nabla}_\phi J(\theta, \phi_k) \rangle \tag{53}$$

$$\geq \langle \frac{\partial J(\theta, \phi_k)}{\partial \phi}, \frac{\partial \hat{J}(\theta, \phi_k)}{\partial \phi} \rangle - \left\| \frac{\partial J(\theta, \phi_k)}{\partial \phi} \right\| \left\| \frac{\partial \hat{J}(\theta, \phi_k)}{\partial \phi} - \hat{\nabla}_\phi J(\theta, \phi_k) \right\| \tag{54}$$

$$\geq \langle \frac{\partial J(\theta, \phi_k)}{\partial \phi}, \frac{\partial \hat{J}(\theta, \phi_k)}{\partial \phi} \rangle - T \tilde{C} h \left\| \frac{\partial J(\theta, \phi_k)}{\partial \phi} \right\|, \tag{55}$$

where the last step follows from Eq. (50).

Thus, we have

$$J(\theta, \phi_{k+1}) \geq J(\theta, \phi_k) + \eta_\phi \langle \frac{\partial J(\theta, \phi_k)}{\partial \phi}, \hat{\nabla}_\phi J(\theta, \phi_k) \rangle - \frac{L_\phi}{2} \eta_\phi^2 \| \hat{\nabla}_\phi J(\theta, \phi_k) \|^2 \tag{56}$$

$$\geq J(\theta, \phi_k) + \eta_\theta \langle \frac{\partial J(\theta, \phi_k)}{\partial \phi}, \frac{\partial \hat{J}(\theta, \phi_k)}{\partial \phi} \rangle - \eta_\theta T \tilde{C} h \left\| \frac{\partial J(\theta, \phi_k)}{\partial \phi} \right\| \tag{57}$$

$$- \frac{3L_\phi}{2} \eta_\phi^2 \left( \left\| \frac{\partial J(\theta, \phi_k)}{\partial \phi} \right\|^2 + \left\| \frac{\partial J}{\partial \phi} - \frac{\partial \hat{J}}{\partial \phi} \right\|^2 + T^2 \tilde{C}^2 h^2 \right), \tag{58}$$

where the last inequality follows from the Cauchy-Schwarz inequality: $\|\hat{\nabla}_\phi J\|^2 = \|\frac{\partial J}{\partial \phi} + (\frac{\partial \hat{J}}{\partial \phi} - \frac{\partial J}{\partial \phi}) + \left(\hat{\nabla}_\phi J - \frac{\partial \hat{J}}{\partial \phi}\right)\|^2 \leq$
$3(\|\frac{\partial J}{\partial \phi}\|^2 + \|\hat{\nabla}_\phi J - \frac{\partial \hat{J}}{\partial \phi}\|^2 + \|\frac{\partial \hat{J}}{\partial \phi} - \frac{\partial J}{\partial \phi}\|^2)$ and Eq. (50).

Taking expectation $\mathbb{E}[\cdot \mid \mathcal{F}_k]$ on both sides, we have

$$\mathbb{E}\left[J(\theta, \phi_{k+1}) \mid \mathcal{F}_k\right] \tag{59}$$

$$\geq J(\theta, \phi_k) + \eta_\phi \left\|\frac{\partial J(\theta, \phi_k)}{\partial \phi}\right\|^2 - \eta_\theta T\tilde{C}h \left\|\frac{\partial J(\theta, \phi_k)}{\partial \phi}\right\| - \frac{3L_\phi}{2}\eta_\phi^2 \left(\left\|\frac{\partial J(\theta, \phi_k)}{\partial \phi}\right\|^2 + (\sigma')^2 + T^2\tilde{C}^2h^2\right) \tag{60}$$

$$= J(\theta, \phi_k) + \left(\eta_\phi - \frac{3L_\phi\eta_\phi^2}{2}\right)\left\|\frac{\partial J(\theta, \phi_k)}{\partial \phi}\right\|^2 - \eta_\theta T\tilde{C}h\left\|\frac{\partial J(\theta, \phi_k)}{\partial \phi}\right\| - \frac{3L_\phi\eta_\phi^2\left((\sigma')^2 + T^2\tilde{C}^2h^2\right)}{2} \tag{61}$$

$$\geq J(\theta, \phi_k) + \frac{1}{2}\left(\eta_\phi - \frac{3L_\phi\eta_\phi^2}{2}\right)\left\|\frac{\partial J(\theta, \phi_k)}{\partial \phi}\right\|^2 - \frac{3L_\phi\eta_\phi^2\left((\sigma')^2 + T^2\tilde{C}^2h^2\right)}{2}, \tag{62}$$

where the last step follows from the assumption that $N \geq \frac{2\eta_\theta T\tilde{C}}{\left(\eta_\phi - \frac{3L_\phi\eta_\phi^2}{2}\right)\|\partial J(\theta, \phi)/\partial \phi\|}$.

Rearranging the terms, we have

$$\frac{1}{2}\left(\eta_\phi - \frac{3L_\phi\eta_\phi^2}{2}\right)\left\|\frac{\partial J(\theta, \phi_k)}{\partial \phi}\right\|^2 \leq \mathbb{E}\left[J(\theta, \phi_{k+1}) - J(\theta, \phi_k) \mid \mathcal{F}_k\right] + \frac{3L_\phi\eta_\phi^2\left((\sigma')^2 + T^2\tilde{C}^2h^2\right)}{2}, \tag{63}$$

which completes the proof of Lemma A.7. $\qquad\qquad\square$

*Proof of Theorem 4.2.* We can decompose the $L_2$-norm of gradient w.r.t. $\theta$ and $\phi$ at the $k$-th training iteration into:

$$\|\nabla_{\theta,\phi}J(\theta_k, \phi_k)\|^2 \tag{64}$$

$$= \|\nabla_\theta J(\theta_k, \phi_k)\|^2 + \left\|\frac{\partial J(\theta_k, \phi_k)}{\partial \phi}\right\|^2 \tag{65}$$

$$\leq \|\nabla_\theta J(\theta_k, \phi_k)\|^2 + 2\left\|\frac{\partial J(\theta_{k+1}, \phi_k)}{\partial \phi}\right\|^2 + 2\left\|\frac{\partial J(\theta_k, \phi_k)}{\partial \phi} - \frac{\partial J(\theta_{k+1}, \phi_k)}{\partial \phi}\right\|^2 \tag{66}$$

$$\leq \|\nabla_\theta J(\theta_k, \phi_k)\|^2 + 2\left\|\frac{\partial J(\theta_{k+1}, \phi_k)}{\partial \phi}\right\|^2 + 2L_J^2\|\theta_{k+1} - \theta_k\| \tag{67}$$

$$= \|\nabla_\theta J(\theta_k, \phi_k)\|^2 + 2\left\|\frac{\partial J(\theta_{k+1}, \phi_k)}{\partial \phi}\right\|^2 + 2L_J^2\eta_\theta^2 \cdot \|\hat{\nabla}_\theta J(\theta_k, \phi_k)\|^2 \tag{68}$$

$$\leq (1 + 4L_J^2\eta_\theta^2)\|\nabla_\theta J(\theta_k, \phi_k)\|^2 + 2\left\|\frac{\partial J(\theta_{k+1}, \phi_k)}{\partial \phi}\right\|^2 + 4L_J^2\eta_\theta^2 \cdot \left\|\hat{\nabla}_\theta J(\theta_k, \phi_k) - \nabla_\theta J(\theta_k, \phi_k)\right\|^2, \tag{69}$$

where the first inequality follows from the Cauchy-Schwarz inequality, the second inequality follows from Assumption A.2, the last inequality follows from the Cauchy-Schwarz inequality.

We multiply $C = \min\left\{\frac{1}{2}\left(\eta_\phi - \frac{3L_\phi\eta_\phi^2}{2}\right), \eta_\theta - L_\theta\eta_\theta^2\right\} > 0$ on both sides and take a total expectation on both sides, then under Assumption A.4, we have

$$C \cdot \mathbb{E}\left[\|\nabla_{\theta,\phi}J(\theta_k, \phi_k)\|^2\right]$$

$$\leq (1 + 4L_J^2\eta_\theta^2)C \cdot \mathbb{E}\left[\|\nabla_\theta J(\theta_k, \phi_k)\|^2\right] + 2C \cdot \mathbb{E}\left[\left\|\frac{\partial J(\theta_{k+1}, \phi_k)}{\partial \phi}\right\|^2\right] + 4L_J^2\eta_\theta^2C \cdot \left\|\hat{\nabla}_\theta J(\theta_k, \phi_k) - \nabla_\theta J(\theta_k, \phi_k)\right\|^2$$

$$\leq 2(1 + 4L_J^2\eta_\theta^2)(\eta_\theta - L_\theta\eta_\theta^2)\mathbb{E}\left[\|\nabla_\theta J(\theta_k, \phi_k)\|^2\right] + 2(1 + 4L_J^2\eta_\theta^2)\left(\eta_\phi - \frac{3L_\phi\eta_\phi^2}{2}\right)\mathbb{E}\left[\left\|\frac{\partial J(\theta_{k+1}, \phi_k)}{\partial \phi}\right\|^2\right] + 4L_J^2\eta_\theta^2C\sigma^2.$$

Then, we use Lemma A.6 with and Lemma A.7 to further bound the right hand side of the above inequality, and compute the average from $k = 1$ to $K$:

$$C \cdot \frac{1}{K} \sum_{k=0}^{K-1} \mathbb{E}\left[\|\nabla_{\theta,\phi} J(\theta_k, \phi_k)\|^2\right]$$

$$\leq \frac{2(1 + 4L_J^2 \eta_\theta^2)}{K} \sum_{k=0}^{K-1} \mathbb{E}\left[J(\theta_{k+1}, \phi_k) - J(\theta_k, \phi_k)\right] + \frac{2(1 + 4L_J^2 \eta_\theta^2)}{K} \sum_{k=0}^{K-1} \mathbb{E}\left[J(\theta_{k+1}, \phi_{k+1}) - J(\theta_{k+1}, \phi_k)\right]$$

$$+ 2(1 + 4L_J^2 \eta_\theta^2) L_\theta \eta_\theta^2 \sigma^2 + \frac{6(1 + 4L_J^2 \eta_\theta^2) L_\phi \eta_\phi^2 \left((\sigma')^2 + T^2 \tilde{C}^2 h^2\right)}{2} + 4L_J^2 \eta_\theta^2 C \sigma^2$$

$$= \frac{2(1 + 4L_J^2 \eta_\theta^2)}{K} \sum_{k=0}^{K-1} \mathbb{E}\left[J(\theta_{k+1}, \phi_{k+1}) - J(\theta_k, \phi_k)\right]$$

$$+ 2(1 + 4L_J^2 \eta_\theta^2) L_\theta \eta_\theta^2 \sigma^2 + \frac{6(1 + 4L_J^2 \eta_\theta^2) L_\phi \eta_\phi^2 \left((\sigma')^2 + T^2 \tilde{C}^2 h^2\right)}{2} + 4L_J^2 \eta_\theta^2 C \sigma^2$$

$$= \frac{2(1 + 4L_J^2 \eta_\theta^2)}{K} \left(J(\theta_K, \phi_K) - J(\theta_0, \phi_0)\right)$$

$$+ 2(1 + 4L_J^2 \eta_\theta^2) L_\theta \eta_\theta^2 \sigma^2 + \frac{6(1 + 4L_J^2 \eta_\theta^2) L_\phi \eta_\phi^2 \left((\sigma')^2 + T^2 \tilde{C}^2 h^2\right)}{2} + 4L_J^2 \eta_\theta^2 C \sigma^2.$$

Then, we have:

$$C \cdot \frac{1}{K} \sum_{k=0}^{K-1} \mathbb{E}\left[\|\nabla_{\theta,\phi} J(\theta_k, \phi_k)\|^2\right]$$

$$\leq \frac{2(1 + 4L_J^2 \eta_\theta^2)}{K} \left(J(\theta^*, \phi^*) - J(\theta_0, \phi_0)\right)$$

$$+ 2(1 + 4L_J^2 \eta_\theta^2) L_\theta \eta_\theta^2 \sigma^2 + \frac{6(1 + 4L_J^2 \eta_\theta^2) L_\phi \eta_\phi^2 \left((\sigma')^2 + T^2 \tilde{C}^2 h^2\right)}{2} + 4L_J^2 \eta_\theta^2 C \sigma^2.$$

We set $\eta_\theta = \frac{\alpha_\theta}{\sqrt{K}}, \eta_\phi = \frac{\alpha_\phi}{\sqrt{K}}$ where $\alpha_\theta, \alpha_\phi$ denote some constants, and denote $C' = \min\left\{\frac{1}{2}\left(\alpha_\phi - \frac{L_\phi \alpha_\phi}{2\sqrt{K}}\right), \alpha_\theta - \frac{L_\theta \alpha_\theta}{\sqrt{K}}\right\}$, then we have

$$\frac{1}{K} \sum_{k=0}^{K-1} \mathbb{E}\left[\|\nabla_{\theta,\phi} J(\theta_k, \phi_k)\|^2\right] \tag{70}$$

$$\leq \left(J(\theta^*, \phi^*) - J(\theta_0, \phi_0) + L_\theta \alpha_\theta^2 \sigma^2 + \frac{3L_\phi \alpha_\phi^2 \left((\sigma')^2 + T^2 \tilde{C}^2 h^2\right)}{2}\right) \cdot \frac{2(1 + 4L_J^2 \eta_\theta^2)}{\sqrt{K} C'} + \frac{4L_J^2 \alpha_\theta^2 \sigma^2}{K} \tag{71}$$

$$= \tilde{\mathcal{O}}\left(\frac{\kappa}{\sqrt{K} C'} + \frac{L_J^2 \alpha_\theta^2 \sigma^2}{K}\right), \tag{72}$$

where

$$\kappa := J(\theta^*, \phi^*) - J(\theta_0, \phi_0) + L_\theta \alpha_\theta^2 \sigma^2 + \frac{3L_\phi \alpha_\phi^2 \left((\sigma')^2 + T^2 \tilde{C}^2 h^2\right)}{2} \tag{73}$$

denotes a constant with the optimal parameters $\theta^*, \phi^* = \arg\max_{\theta,\phi} J(\theta, \phi)$ and the inital parameters $\theta_0, \phi_0$, the approximation in the last step follows from the fact that $1 + 4L_J^2 \eta_\theta^2 = 1 + \frac{4L_J^2 \alpha_\theta^2}{K} \to 1$ as $K \to \infty$ (the number of optimization iterations is usually sufficiently large).

This completes the proof of Theorem 4.2. $\qquad\qquad\qquad\qquad\qquad\qquad\qquad\qquad\qquad\qquad\qquad\qquad\qquad\quad \square$

## A.2. Proof of Theorem 4.4

*Proof of Theorem 4.4.* By definition of KL divergence:

$$\text{KL}\left(\mathbb{P}(\cdot \mid x, \pi) \| \mathbb{P}(\cdot \mid x, \pi_{\text{ref}})\right) = \mathbb{E}_\pi \left[ \log \frac{\mathbb{P}(a_{0:T-1}^\pi \mid X_0^x, \pi)}{\mathbb{P}(a_{0:T-1}^\pi \mid x, \pi_{\text{ref}})} \middle| X_0^x \right]. \tag{74}$$

For the learning policy $\pi$, the action sequence probability can be derived by marginalizing over the latent state trajectories. The distribution of trajectory $\tau = \{X_0^x, a_0^\pi, X_1^\pi, \ldots, X_{T-1}^\pi, a_{T-1}^\pi, X_T^\pi\}$ given the initial state $X_0^x$ under $\pi$ is:

$$\mathbb{P}(\tau) = \prod_{t=0}^{T-1} \pi(a_t^\pi \mid X_t^\pi) \cdot \prod_{t=0}^{T-1} \delta\left(X_{t+1}^\pi - \text{Flow}_{t \to t+1}(X_t^\pi, a_t^\pi)\right), \tag{75}$$

where $\text{Flow}_{t \to t+1}(X_t^\pi, a_t^\pi)$ is the deterministic flow from time $t$ to $t+1$ defined by the ODE Eq. (1), and $\delta(\cdot)$ denotes the Dirac delta function representing deterministic transitions.

Integrating over the latent states yields:

$$\begin{aligned}
\mathbb{P}(a_{0:T-1}^\pi \mid X_0^x, \pi) &= \int \prod_{t=0}^{T-1} \pi(a_t^\pi \mid X_t) \cdot \prod_{t=0}^{T-1} \delta\left(X_{t+1} - \text{Flow}_{t \to t+1}(X_t, a_t^\pi)\right) \, dX_{1:T} \\
&= \prod_{t=0}^{T-1} \pi(a_t^\pi \mid X_t^\pi),
\end{aligned} \tag{76}$$

where $X_0^x$ is a fixed initial state, and for $t \geq 1$, $X_t^\pi = \text{Flow}_{0 \to t}(X_0^x, a_{0:t-1}^\pi)$ is uniquely determined by the initial state and previous actions.

For the reference policy $\pi_{\text{ref}}$, which operates directly on original states $\tilde{s}_t = (x, a_0, \ldots, a_{t-1})$, we have:

$$\mathbb{P}(a_{0:T-1} \mid x, \pi_{\text{ref}}) = \prod_{t=0}^{T-1} \pi_{\text{ref}}(a_t \mid \tilde{s}_t), \tag{77}$$

where $(x, a_{<t}^\pi) = h(X_t^\pi)$ is obtained by applying the mapping $h$ to the latent state $X_t^\pi$ (from Definition 4.3). Since the dynamics are deterministic, $\tilde{s}_t$ is also uniquely determined by the initial state and previous actions.

Substituting Eq. (76) and Eq. (77) into Eq. (74), we obtain:

$$\begin{aligned}
\text{KL}\left(\mathbb{P}(\cdot \mid x, \pi) \| \mathbb{P}(\cdot \mid x, \pi_{\text{ref}})\right) &= \mathbb{E}_\pi \left[ \sum_{t=0}^{T-1} \log \frac{\pi(a_t^\pi \mid X_t^\pi)}{\pi_{\text{ref}}(a_t^\pi \mid \tilde{s}_t)} \middle| X_0^x \right] \tag{78} \\
&= \mathbb{E}_\pi \left[ \sum_{t=0}^{T-1} \log \frac{\pi(a_t^\pi \mid X_t^\pi)}{\pi_{\text{ref}}(a_t^\pi \mid h(X_t^\pi))} \middle| X_0^x \right] \tag{79} \\
&= \mathbb{E}_\pi \left[ \sum_{t=0}^{T-1} \mathbb{E}_\pi \left[ \log \frac{\pi(a_t^\pi \mid X_t^\pi)}{\pi_{\text{ref}}(a_t^\pi \mid h(X_t^\pi))} \middle| X_t^\pi \right] \middle| X_0^x \right] \tag{80} \\
&= \mathbb{E}_\pi \left[ \sum_{t=0}^{T-1} \text{KL}\left(\pi(\cdot \mid X_t^\pi) \| \pi_{\text{ref}}(\cdot \mid h(X_t^\pi))\right) \middle| X_0^x \right], \tag{81}
\end{aligned}$$

where the outer expectation is taken over the trajectory $(X_0^\pi, a_0^\pi, X_1^\pi, \ldots, a_{T-1}^\pi, X_T^\pi)$ generated by the policy $\pi$ given the inital state $X_0^x$, the second step follows from Definition 4.3, such that: $\tilde{s}_t = (x, a_{<t}^\pi) = h(X_t^\pi)$, the third step follows from tower property, the last step follows from the definition of Kullback-Leibler divergence.

The last equality follows from the definition of KL divergence:

$$\text{KL}\left(\pi(\cdot \mid X_t^\pi) \| \pi_{\text{ref}}(\cdot \mid s_t)\right) = \mathbb{E}_{a_t^\pi \sim \pi(\cdot \mid X_t^\pi)} \left[ \log \frac{\pi(a_t^\pi \mid X_t^\pi)}{\pi_{\text{ref}}(a_t^\pi \mid h(X_t^\pi))} \right]. \tag{82}$$

This completes the proof. $\square$

### A.3. Proof of Theorem 4.5

*Proof of Theorem 4.5.* The objective of policy optimization with KL regularization:

$$J_{\mathrm{KL}}(\pi) = V^\pi(\mu) - \beta \mathbb{E}_{x \sim \mu} \left[ \mathrm{KL} \left( \mathbb{P}(\cdot \mid x, \pi) \parallel \mathbb{P}(\cdot \mid x, \pi_{\mathrm{ref}}) \right) \right]. \tag{83}$$

We denote the optimal policy on KL-regularized objective as: $\hat{\pi} = \arg\max_\pi J_{\mathrm{KL}}(\pi)$, and the optimal policy on value function as: $\pi^* = \arg\max_\pi V^\pi(\mu)$.

Following from the fact that $J_{\mathrm{KL}}(\hat{\pi}) \geq J_{\mathrm{KL}}(\pi^*)$, we have

$$V^{\hat{\pi}}(\mu) - \beta \mathbb{E}_{x \sim \mu} \left[ \mathrm{KL} \left( \mathbb{P}(\cdot \mid x, \hat{\pi}) \parallel \mathbb{P}(\cdot \mid x, \pi_{\mathrm{ref}}) \right) \right] \geq V^{\pi^*}(\mu) - \beta \mathbb{E}_{x \sim \mu} \left[ \mathrm{KL} \left( \mathbb{P}(\cdot \mid x, \pi^*) \parallel \mathbb{P}(\cdot \mid x, \pi_{\mathrm{ref}}) \right) \right]. \tag{84}$$

Rearranging the terms and using that fact that $\mathrm{KL} \geq 0$,

$$V^{\hat{\pi}}(\mu) \geq V^{\pi^*}(\mu) - \beta \mathbb{E}_{x \sim \mu} \left[ \mathrm{KL} \left( \mathbb{P}(\cdot \mid x, \pi^*) \parallel \mathbb{P}(\cdot \mid x, \pi_{\mathrm{ref}}) \right) \right] + \beta \mathbb{E}_{x \sim \mu} \left[ \mathrm{KL} \left( \mathbb{P}(\cdot \mid x, \hat{\pi}) \parallel \mathbb{P}(\cdot \mid x, \pi_{\mathrm{ref}}) \right) \right] \tag{85}$$

$$\geq V^{\pi^*}(\mu) - \beta \mathbb{E}_{x \sim \mu} \left[ \mathrm{KL} \left( \mathbb{P}(\cdot \mid x, \pi^*) \parallel \mathbb{P}(\cdot \mid x, \pi_{\mathrm{ref}}) \right) \right]. \tag{86}$$

Thus, we have

$$\begin{aligned} V^{\pi^*}(\mu) - V^{\hat{\pi}}(\mu) &\leq \beta \mathbb{E}_{x \sim \mu} \left[ \mathrm{KL} \left( \mathbb{P}(\cdot \mid x, \pi^*) \parallel \mathbb{P}(\cdot \mid x, \pi_{\mathrm{ref}}) \right) \right] \\ &= \beta \mathbb{E}_{x \sim \mu} \mathbb{E}_{\pi^*} \left[ \sum_{t=0}^{T-1} \mathrm{KL} \left( \pi^*(\cdot \mid X_t^{\pi^*}) \parallel \pi_{\mathrm{ref}}(\cdot \mid h(X_t^{\pi^*})) \right) \Big| X_0^x \right]. \end{aligned} \tag{87}$$

Using the performance difference lemma (PDL) and the fact that $\mathbb{E}_{a \sim \pi(\cdot \mid X_t^{\hat{\pi}})}[A^\pi(t, X_t^{\hat{\pi}}, a)] = 0$, we have

$$\begin{aligned} V^{\hat{\pi}}(\mu) - V^\pi(\mu) &= \mathbb{E}_{x \sim \mu} \mathbb{E}_{\hat{\pi}} \left[ \sum_{t=0}^{T-1} \gamma^t A^\pi(t, X_t^{\hat{\pi}}, a_t^{\hat{\pi}}) \mid X_0^x \right] \\ &= \sum_{t=0}^{T-1} \gamma^t \mathbb{E}_{x \sim \mu} \mathbb{E}_{\hat{\pi}} \left[ \mathbb{E}_{a \sim \hat{\pi}(\cdot \mid X_t^{\hat{\pi}})}[A^\pi(t, X_t^{\hat{\pi}}, a)] - \mathbb{E}_{a \sim \pi(\cdot \mid X_t^{\hat{\pi}})}[A^\pi(t, X_t^{\hat{\pi}}, a)] \mid X_0^x \right] \\ &= \sum_{t=0}^{T-1} \gamma^t \mathbb{E}_{x \sim \mu} \mathbb{E}_{\hat{\pi}} \left[ \int_{\mathcal{A}} \left( \hat{\pi}(a \mid X_t^{\hat{\pi}}) - \pi(a \mid X_t^{\hat{\pi}}) \right) A^\pi(t, X_t^{\hat{\pi}}, a) \mathrm{d}a \mid X_0^x \right] \\ &\leq \sum_{t=0}^{T-1} \gamma^t \mathbb{E}_{x \sim \mu} \mathbb{E}_{\hat{\pi}} \left[ \left\| \hat{\pi}(\cdot \mid X_t^{\hat{\pi}}) - \pi(\cdot \mid X_t^{\hat{\pi}}) \right\|_1 \cdot \max_{a \in \mathcal{A}} |A^\pi(t, X_t^{\hat{\pi}}, a)| \mid X_0^x \right], \end{aligned} \tag{88}$$

where $\| \cdot \|_1$ denotes the $L_1$-norm.

Using the assumption $r \in [0, 1]$, we have that $\max_{a \in \mathcal{A}} |A^\pi(t, X_t^{\hat{\pi}}, a)| \leq 1/(1 - \gamma)$. Then, following from Pinsker's inequality: $\left\| \hat{\pi}(\cdot \mid X_t^{\hat{\pi}}) - \pi(\cdot \mid X_t^{\hat{\pi}}) \right\|_1 \leq \sqrt{2 \mathrm{KL} \left( \hat{\pi}(\cdot \mid X_t^{\hat{\pi}}) \| \pi(\cdot \mid X_t^{\hat{\pi}}) \right)}$ and the fact that $\gamma^t \leq 1$, we have

$$\text{RHS of Eq. (88)} \leq \frac{1}{1 - \gamma} \sum_{t=0}^{T-1} \gamma^t \mathbb{E}_{x \sim \mu} \mathbb{E}_{\hat{\pi}} \left[ \sqrt{2 \mathrm{KL} \left( \hat{\pi}(\cdot \mid X_t^{\hat{\pi}}) \parallel \pi(\cdot \mid X_t^{\hat{\pi}}) \right)} \mid X_0^x \right] \tag{89}$$

$$\leq \frac{1}{1 - \gamma} \sum_{t=0}^{T-1} \sqrt{2 \mathbb{E}_{x \sim \mu} \mathbb{E}_{\hat{\pi}} \left[ \mathrm{KL} \left( \hat{\pi}(\cdot \mid X_t^{\hat{\pi}}) \parallel \pi(\cdot \mid X_t^{\hat{\pi}}) \right) \mid X_0^x \right]}, \tag{90}$$

where the second inequality follows from Jensen's inequality.

Thus, we have

$$V^{\hat{\pi}}(\mu) - V^\pi(\mu) \leq \frac{1}{1 - \gamma} \sum_{t=0}^{T-1} \sqrt{2 \mathbb{E}_{x \sim \mu} \mathbb{E}_{\hat{\pi}} \left[ \mathrm{KL} \left( \hat{\pi}(\cdot \mid X_t^{\hat{\pi}}) \parallel \pi(\cdot \mid X_t^{\hat{\pi}}) \right) \mid X_0^x \right]}. \tag{91}$$

Combining Eq. (87) and Eq. (91), we have

$$
V^{\pi^*}(\mu) - V^{\pi}(\mu)
$$

$$
\leq \beta \mathbb{E}_{x\sim\mu} \mathbb{E}_{\pi^*} \left[ \sum_{t=0}^{T-1} \mathrm{KL}\left( \pi^*(\cdot \mid X_t^{\pi^*}) \| \pi_{\mathrm{ref}}(\cdot \mid h(X_t^{\pi^*})) \right) \Big| X_0^x \right] + \frac{1}{1-\gamma} \sum_{t=0}^{T-1} \sqrt{ 2\mathbb{E}_{x\sim\mu} \mathbb{E}_{\hat{\pi}} \big[ \mathrm{KL}\left( \hat{\pi}(\cdot \mid X_t^{\hat{\pi}}) \| \pi(\cdot \mid X_t^{\hat{\pi}}) \right) \mid X_0^x \big] }
$$

$$
= \sum_{t=0}^{T-1} \left( \beta \mathbb{E}_{x\sim\mu} \mathbb{E}_{\pi^*} \left[ \mathrm{KL}\left( \pi^*(\cdot \mid X_t^{\pi^*}) \| \pi_{\mathrm{ref}}(\cdot \mid h(X_t^{\pi^*})) \right) \Big| X_0^x \right] + \frac{1}{1-\gamma} \sqrt{ 2\mathbb{E}_{x\sim\mu} \mathbb{E}_{\hat{\pi}} \left[ \mathrm{KL}\left( \hat{\pi}(\cdot \mid X_t^{\hat{\pi}}) \| \pi(\cdot \mid X_t^{\hat{\pi}}) \right) | X_0^x \right] } \right),
$$

which completes the proof of Theorem 4.5.

$\square$

# B. Experimental Setup in Details

We summarize the detailed architecture comparisons between our CT-LLM structure and the Transformer baseline (Vaswani et al., 2017) in Table 6. We meticulously aligned the key hyperparameters (e.g., vocabulary size, layer normalization, activation function, parameter size) between CT-LLM and the Transformer baseline. Regarding the differences in Attention Heads and Position/Time Embedding:

- Our ODE-based architecture inherently contains no attention modules, thus naturally eliminating the need for both attention heads and relative position embedding like RoPE (Su et al., 2024).

- For the Transformer baseline, the configurations of RoPE and attention heads follow advanced LLaMA (Touvron et al., 2023) or Qwen (Bai et al., 2023) models, which we empirically verified to be optimal (e.g., outperforming Sinusoidal alternatives for Position Embedding (Vaswani et al., 2017)) under our experimental settings.

*Table 6.* Model architectures of CT-LLM and the baseline Transformer.

| Hyperparameter | CT-LLM | Transformer (Baseline) |
|---|---:|---:|
| Layer ($L_n + L_p$) | 12 (ODE Nets) + 6 (Policy Nets) | 18 |
| Hidden Dimension ($d$) | 2,048 | 2,048 |
| FFN Dimension ($d \cdot r_n, d \cdot r_p$) | 8,192 | 8,192 |
| Attention Heads | - | 16 (Q) + 8 (K/V) |
| Activation Function | SwiGLU (ODE Nets), GeGLU (Policy Nets) | SwiGLU |
| Vocabulary Size ($|\mathcal{A}|$) | 128,256 | 128,256 |
| Positon/Time Embedding ($\mathcal{E}_t$) | Sinusoidal ($\theta = 500,000$) | RoPE ($\theta = 500,000$) |
| Layer Normalization | RMSNorm | RMSNorm |
| **Parameter Size** | $\approx 1.2\mathrm{B}$ | $\approx 1.7\mathrm{B}$ |

*Table 7.* Training hyperparameters for the two-phase optimization. The supervised phase prepares the policy network, while the RL phase jointly fine-tunes the policy and the ODE dynamics.

| Hyperparameter | Value | Description |
|---|---|---|
| *Supervised pre-training phase* | | |
| Optimizer | AdamW | $\beta_1 = 0.9, \; \beta_2 = 0.999$ |
| Learning rate | $5 \times 10^{-5}$ | Linear warmup + cosine decay |
| Batch size | 32 | Number of prompts |
| Epochs | 3 | Over the instruction dataset |
| *Reinforcement learning phase* | | |
| Optimizer | AdamW | Same $\beta_1, \beta_2$ |
| Learning rate $(\eta_\theta, \eta_\phi)$ | $1 \times 10^{-4}$ | Linear warmup + cosine decay |
| Batch size | 32 | Number of prompts |
| Sequence length $(T)$ | 2048 | Maximum generation steps |
| Number of ODE steps $(N)$ | 10 | ODE solver steps per token interval |
| Discount factor $(\gamma)$ | 0.99 | Reward discount |
| KL coefficient $(\beta)$ | $0.1 \rightarrow 0.01$ | Linear annealing |
| Gradient clipping | 1.0 | Global norm |
| Training iterations $(K)$ | 50,000 | For each experiment |
| *Hardware & software* | | |
| GPUs | $8 \times$ NVIDIA A100 80GB | Distributed training |
| Framework | PyTorch 2.1 + DeepSpeed | ZeRO-Stage 2 |
| ODE library | torchdiffeq | Adjoint-mode backpropagation |

