# OpenReview forum: "Text Generation as Continuous Latent Dynamics via Reinforcement Learning"
_ICML.cc/2026/Conference — ICML 2026 regular_

### Official Review · Reviewer_mhqk · 2026-03-12

**Soundness:** 2
**Presentation:** 2
**Significance:** 2
**Originality:** 3
**Overall Recommendation:** 4
**Confidence:** 2

**Summary:**

This paper proposes modeling text generation as a continuous-time latent dynamical process, where token generation is formulated as a Markov Decision Process, and the internal latent state evolves according to an ODE. The proposed Continuous-Time Latent Language Model (CT-LLM) jointly learns a token generation policy and latent dynamics using reinforcement learning, combining a reward objective with knowledge distillation from a pretrained teacher model.

**Compliance With Llm Reviewing Policy:**

Affirmed.

**Final Justification:**

The new results on standard long-context benchmarks are helpful and make the empirical claims much clearer. If the method can consistently demonstrate both improved effectiveness in long-context settings and higher computational efficiency than standard Transformers with full self-attention, it would constitute a highly impactful and pioneering contribution with broad implications for future research and practical applications.

The authors have addressed some of my concerns, and I have raised my score to 4.

**Key Questions For Authors:**

1. To what extent do the reported improvements come from the proposed ODE-based dynamics rather than the knowledge distillation from the teacher model?

2. Since RL training is often unstable, does the proposed framework introduce additional training difficulty or computational burden?

**Limitations:**

No limitation mentioned.

The paper could be further improved by (1) providing more detailed ablation studies to better isolate the contribution of the proposed ODE dynamics, (2) expanding the empirical evaluation to a broader set of benchmarks to demonstrate generalization, and (3) including visualizations or interpretability analyses to better illustrate the behavior of the proposed continuous latent dynamics.

**Strengths And Weaknesses:**

**Strengths**

- The paper proposes an interesting continuous-time perspective on text generation by introducing neural ODE-based latent dynamics.
- The work provides theoretical analysis, including KL divergence decomposition and convergence guarantees for the optimization procedure.
- The proposed model demonstrates encouraging parameter efficiency compared to large teacher models.



**Weaknesses**

- Figure 1 could benefit from clearer explanations. In particular, it would be helpful to explicitly illustrate how the proposed framework differs from the standard LLM decoding process.
- Section 4.3 would be easier to understand with additional architectural diagrams or visual illustrations.
- The practical efficiency advantages over standard architectures remain unclear (the same parameter size).
- The empirical evaluation is relatively limited and does not fully demonstrate generalization across a wider range of tasks.

---

> ### Author Rebuttal · Authors · 2026-03-31
>
> Thank you for taking the time to read our paper and giving constructive comments. We hope your concerns can be addressed by our point-to-point responses.
>
> > **Q1**: To what extent do the reported improvements come from the proposed ODE-based dynamics rather than the knowledge distillation from the teacher model?
>
> The performance gain stems primarily from the expressiveness of the ODE dynamics, while Knowledge Distillation (KD) serves as a stability regularizer.
>
> #### 1. Empirical Evidence: KD is Necessary but Insufficient
> To isolate the contribution of the ODE dynamics from the KD signal, we conducted an ablation study on the KD coefficient $\beta$ (Eq. 4 in the paper) and compared our ODE-based model against a discrete-time baseline trained with the *identical* KD objective. The table below shows the ROUGE-L scores of our model (CT-LLM) and a discrete-time Transformer baseline against $\beta$ from 0 to 1.0 on AlphaEval 2.
>
> | $\beta$ | 0 | $10^{-4}$ | $10^{-3}$ | 0.01 | 0.05 | 0.1 | 0.2 | 0.5 | 1.0 |
> | :--- | :--- | :--- | :--- | :--- | :--- | :--- | :--- | :--- | :--- |
> | **CT-LLM (Ours)** | 30.73 | 32.76 | 34.58 | 35.76 | **36.75** | 36.24 | 34.34 | 33.91 | 32.78 |
> | **Transformer Baseline** | 29.50 | 30.52 | 30.34 | 31.34 | 30.65 | 30.03 | - | - | - |
>
> The ablation study on the KD coefficient $\beta$ (Table above) reveals two key points:
>
> - The ODE dynamics are essential: Even without KD ($\beta=0$), our model (30.73) significantly outperforms the discrete baseline (**29.50**). This proves the ODE formulation has a superior inductive bias.
> - KD alone is insufficient: When trained with the optimal KD signal (e.g., $\beta=0.01$), the Transformer baseline only reaches 31.34, far below our performance (35.76). This shows that the performance leap requires the ODE formulation; KD cannot compensate for a weaker architecture.
>
> #### 2. Theoretical Perspective: Convergence
> Our objective (Eq. 4) is:
> $$ \max_{\pi,b} \mathbb{E} [r^\star(X_{T-1}, a_{T-1})- \beta {\rm KL}(\pi || \pi_{\rm ref}) ] $$
>
> - Role of KD: In our objective, the KD term $-\beta {\rm KL}(\pi || \pi_{\rm ref})$ serves as a *trust region constraint* to prevent policy collapse. It restricts the search space but does not directly improve the reward.
> - Optimization Target: As proven in Thm 4.3 (Convergence), the algorithm converges to a stationary point of the **expected reward** $r^\star$. The ODE dynamics $b_\phi$ provide a richer function space to approximate the optimal value function $V^*$, which is the source of the performance gain.
>
> > **Q2**: Since RL training is often unstable, does the proposed framework introduce additional training difficulty or computational burden?
>
> #### 1. Theoretical Analysis:
> Our framework introduces negligible computational overhead and maintains training stability via efficient gradient computation and a parameter-efficient architecture. We compare single-layer one-iteration complexity ($T$: seq. length, $d$: latent dim, $N$: ODE steps):
>
> | Cmplx (1-layer/iter) | Standard Transformer | Latent ODE (Ours) | Advantage |
> | ------ |------ |------ |------ |
> | **Memory** | $O(T d)$ | $O(d)$ | Decoupled from sequence length. |
> | **Computation** | $O( Td^2 + T^2d)$ | $O(T N d^2)$ | Avoids $O(T^2)$ attention cost (when $T \gg N,d$). |
>
> - Memory Efficiency. The primary concern with ODE solvers is the memory cost of backpropagation. We employ the continuous adjoint method (Lines 12-14 in Alg. 1). This reduces the memory footprint to $O(d)$, where $d$ is the latent dimension. This cost is independent of the number of solver steps $N$ and seq. length $T$, and significantly smaller than standard Transformers.
>
> - Computational Efficiency & Stability. The per-step complexity scales as $O(T N d^2)$. Crucially, we avoid the $O(T^2)$ self-attention cost, making our method efficient for long context. Regarding stability, our bi-level optimization scheme (alternating policy and ODE updates) is theoretically grounded. Thm 4.3 proves it achieves a convergence rate of $\tilde{O}(1/\sqrt{K})$. This indicates that the total computational complexity required to obtain an $\epsilon$-approximate stationary point is $\tilde{O}(\text{1-iter complex} \cdot \epsilon^{-2})$, which is standard for stochastic optimization, confirming practical feasibility.
>
> #### 2. Empirical Study:
> See response to **Q2 of Reviewer kJU2** for discussions on computational efficiency of our method.
>
> We thank you for the valuable comments and will make the following revisions regarding the Weaknesses:
>
> - **Fig. 1:** Redrawn to explicitly highlight differences between our framework and standard LLM decoding.
> - **Sec. 4.3:** Add supplementary architectural diagrams to improve readability.
> -  **Efficiency:** Add both theoretical and empirical analysis on efficiency advantages (See response to **Q2**).
> -  **Evaluation:** Include results on more long-context generation tasks (See response to **Q3 of Reviewer kJU2**).

---

> > ### Author Rebuttal · Reviewer_mhqk · 2026-04-04
> >
> > 1. Why does the distillation effect deteriorate for the Transformer baseline when $\beta$ becomes larger?
> >
> > 2. CT-LLM may achieve higher efficiency than Transformer-based models, but reducing the quadratic attention cost alone does not by itself establish stronger long-context capability. In particular, the reported analysis is based on output-length bins rather than standard long-context benchmarks, so it remains unclear whether CT-LLM truly handles long contexts better or simply scales more efficiently.

---

> > > ### Author Response · Authors · 2026-04-05
> > >
> > > We sincerely appreciate the reviewer’s acknowledgment and the opportunity to address your follow-up question. We hope the following response can clarify your concerns.
> > >
> > > > **Q1:** Why does the distillation effect deteriorate for the Transformer baseline when $\beta$ becomes larger?
> > >
> > > Thank you for this insightful question. The performance drop at high $\beta$ values is fundamentally tied to the nature of our optimization objective (for our model and the Transformer baseline):
> > >
> > > $$\max_{\pi,b} \mathbb{E} [r^\star(X_{T-1}, a_{T-1}) - \beta {\rm KL}(\pi \| \pi_{\rm ref}) ]$$ (Eq. 4 in the paper)
> > >
> > > In this equation, the KL divergence serves as a regularization constraint. When $\beta$ is increased significantly (e.g., $\geq 0.05$ in our experiments), the weight of this regularization term dominates the overall objective function.
> > >
> > > This leads to a state of "constraint domination". In this regime, the optimization process is forced into strict imitation of the teacher policy ($\pi_{\rm ref}$). While moderate distillation helps stabilize training, too large $\beta$ restricts the model's capacity to explore deviations from the teacher's trajectory, and loses its ability to fully optimize the primary reward signal $r^*$. This trade-off between imitation fidelity and reward optimization aligns with observations in prior studies on RL with KL constraints.
> > >
> > > > **Q2:** In particular, the reported analysis is based on output-length bins rather than standard long-context benchmarks, so it remains unclear whether CT-LLM truly handles long contexts better or simply scales more efficiently.
> > >
> > > We sincerely thank the reviewer for the insightful comment. To validate CT-LLM's capacity to enhance long-context handling (beyond mere scaling efficiency), we have evaluated our model on standard long-context benchmarks.
> > >
> > > Specifically, we adopted **LongBench** (Bai et al., 2024), a widely recognized benchmark specifically designed for long-context understanding. Furthermore, to test the stability of logical reasoning over extended sequences, which is crucial for long-context generation, we included **GSM8K** (mathematical reasoning) and **HumanEval** (code generation). These tasks require the model to maintain coherent chain-of-thought (CoT) reasoning traces or structural dependencies over long distances, directly probing the stability of the latent trajectories.
> > >
> > > | Arch. | Training Strategy | Qasper (F1) | 2WikiMultihopQA (F1) | MultiNews (ROUGE-L) | GSM8K (Accuracy) | HumanEval (Pass@1) |
> > > | :--- | :--- | :--- | :--- | :--- | :--- | :--- |
> > > | Transformer (baseline) | RL | 13.23 | 14.75 | 14.45 | 23.43 | 7.56 |
> > > | Transformer (baseline) | RL+KL | 13.65 | 15.28 | 18.25 | 27.62 | 10.23 |
> > > | CT-LLM (ours) | RL | 15.28 | **18.56** | 18.02 | 28.45 | 18.02 |
> > > | CT-LLM (ours) | RL+KL | **16.45** | 17.23 | **22.48** | **34.57** | **23.43** |
> > >
> > > The above results show that our approach, **CT-LLM**, significantly outperforms discrete-time Transformer baselines on these standardized long-context benchmarks:
> > >
> > > - On LongBench, CT-LLM achieves higher F1 scores across single-document QA (Qasper) and multi-document QA (2WikiMultihopQA), and ROUGE-L score on summarization (MultiNews). This confirms our method’s effectiveness in understanding and synthesizing information over long sequences.
> > > - On GSM8K and HumanEval, CT-LLM achieves higher scores. This indicates that our model maintains a smoother and more stable latent trajectory, which effectively preserves long-range dependencies during long reasoning processes.

---

### Official Review · Reviewer_kJU2 · 2026-03-12

**Soundness:** 3
**Presentation:** 3
**Significance:** 2
**Originality:** 2
**Overall Recommendation:** 4
**Confidence:** 1

**Summary:**

This paper introduces a novel text generation framework CT-LLM. It models the text generation as a continuous time latent dynamical process. The idea is to perceive the token generation as a markov decision process while letting the internal state evolve in a low dimensional latent space. That evaluation is achieved via a neural ODE. Training objective utilizes terminal reward with KL regularization. Emprically, the paper evaluates on UltraFeedback binarized and AlpacaEval comparing parameter matched Llama and T5 baselines trained with different variants like PPO, DPO. The authors report that CT-LLM outperforms the baselines. Furthermore, the paper reports better long context robustness and lower inference latency compared to teacher model.

**Compliance With Llm Reviewing Policy:**

Affirmed.

**Final Justification:**

The paper proposes a technically interesting and reasonably original framework for text generation via continuous latent dynamics.

My main concerns were about whether the gains truly come from the continuous-time formulation, whether the efficiency claims are sufficiently compute-matched, and whether the evaluation is broad enough for the paper’s claims. The rebuttal addressed these concerns partially and strengthened the paper through added ablations, compute-matched results, broader evaluations, and clarification of the baseline.

However, I still have some residual uncertainty about the empirical validation and how strong the benchmark comparisons are in absolute terms. So although the rebuttal improved my view, it did not fully resolve my concerns.

Overall, I remain at 4.

**Key Questions For Authors:**

1)Can the authors provide stronger evidence that the gains come specifically from the continuous-time latent dynamics, rather than from other design choices such as teacher-hidden-state initialization, architecture choices, or the joint RL/KL training setup?

2)The scaling analysis is mainly parameter-based. Do the authors have compute-matched comparisons, or a compute-versus-performance analysis, to support stronger efficiency claims?

3) Can the authors provide broader evaluations beyond AlpacaEval-style benchmarks for long-context generation, to better support the paper’s broader claims?

**Limitations:**

yes

**Strengths And Weaknesses:**

Strengths:
1) The paper proposes to use  continuous latent dynamics rather than purely discrete hidden state evolution which is a concrete algorithmic and architectural change instead of a minor training variation.
2) The emprical results show consistent improvement over the baselines across two teacher models. The ablation results show that both RL and KL regularization matter for the performance improvement.
3) The work is reasonably original in the sense of combining ingredients into a new formulation whose motivation is clearly explained and whose emprical results are shown to be nontrivial.

Weaknesses:
1) In the paper the scaling analysis is mainly parameter vs. performance rather than a compute-scaling one. Although Table 3 reports training FLOPs, compute is not matched across methods (or there is not a compute scaling law plot).
2) While the paper emphasizes coherence and long-context generation, the experimental evaluation is concentrated on AlpacaEval-style preference benchmarks. This makes the empirical evidence promising, but somewhat limited for supporting the paper’s broader claims about text generation more generally.
3) The theoretical framing seem interesting. However, its practical implications seem less clear than the experimental claims. For example some assumptions seem useful for analysis itself. The paper can do more to clarify (provide intution about) the assumptions and tell how central they are for implementing the method.
4) It is unclear how much of the improvement comes from continuous time formuation or other architectural/training choices

---

> ### Author Rebuttal · Authors · 2026-03-31
>
> Thank you for your time, constructive comments, and support. We hope our responses below are satisfactory.
>
> > **Q1:** Can the authors provide stronger evidence that the gains come specifically from the continuous-time latent dynamics, rather than from other design choices such as teacher-hidden-state initialization, architecture choices, or the joint RL/KL training setup?
>
> Certainly. To isolate the contribution of continuous-time dynamics, we conducted ablation studies under identical conditions (without teacher-hidden-state initialization).
>
> **Setup:** 1. Models: Our Continuous-time Model v.s. discrete-time Transformer ( $\approx 1.2B$ params). 2. Conditions: Same training data, infrastructure, and no teacher initialization.
>
> **Results:**  The continuous-time model outperforms the discrete baseline across training strategies (pure RL and joint RL+KL) on AlphaEval 2.
>
> | Arch. | Training Strategy | ROUGE-L | Win Rate |
> | :--- | :--- | :--- | :--- |
> | Discrete-Time Transformer | RL | 29.50 | 11.47 |
> | **CT-LLM (Ours)** | **RL** | **30.53** ($\Delta$+1.03) | **12.63** ($\Delta$+1.16) |
> | Discrete-Time Transformer | RL+KL | 32.52 | 11.40 |
> | **CT-LLM (Ours)** | **RL+KL** | **36.46** ($\Delta$+3.94) | **16.56** ($\Delta$+5.16) |
>
> The performance gain comes from the continuous-time latent dynamics. Under identical training (no initialization trick), the continuous-time model outperforms, particularly in the RL+KL setting, where latent dynamics better exploit KL regularization for stable long-range generation.
>
> > **Q2:** The scaling analysis is mainly parameter-based. Do the authors have compute-matched comparisons, or a compute-versus-performance analysis, to support stronger efficiency claims?
>
> Thanks for your suggestion. We compared our CT-LLM with standard discrete-time Transformer baselines by matching total training FLOPs (compute-matched comparisons) rather than parameter counts.
>
> **Setup:** 1. Total training budget: $1.4 \times 10^{20}$ FLOPs. 2. Baselines: 1.2B-sized Transformer with ~40,000 training iterations and  8B-sized Transformer with 6,000 training iterations.
>
> **Results:** CT-LLM achieves superior performance under the same compute constraints.
>
> | Model                     | Params | Train FLOPs       | Training Iterations | AlpacaEval 2 (ROUGE-L) |
> | :------------------------ | :----- | :---------------- | :------------------ | :--------------- |
> | **CT-LLM (ours)**         | 1.2B   | $\approx 1.4\times10^{20}$ | 50,000         | **36.78**        |
> | Transformer Baseline      | 1.2B   | $\geq1.4\times10^{20}$ | 40,000              | 28.50            |
> | Large Transformer         | 8B     | $\geq 1.4\times10^{20}$ | 6,000               | 30.17            |
>
> CT-LLM demonstrates significantly higher **compute efficiency**. Even when competing against a larger 8B model under the same FLOPs budget, our method achieves a substantial gain of **+6.61%** on AlpacaEval 2. This confirms that our architecture learns more effectively per unit of computation. Besides, a theoretical analysis for compute efficiency of our model can refer to Q2 of Reviewer mhqk).
>
> > **Q3:** Can the authors provide broader evaluations beyond AlpacaEval-style benchmarks for long-context generation, to better support the paper’s broader claims?
>
> Certainly. We evaluated on GSM8K (mathematical reasoning) and HumanEval (code generation) , requiring the model to maintain logical consistency on long-context reasoning and generation.
>
> **Setup:** 1. Prompt: 0-shot Chain-of-Thought (CoT). Reward: A rule-based reward function where the model scores 1.0 for the correct final answer (or all test cases passed for code) and 0.0 for failure.
>
> **Results:** CT-LLM shows significant improvements in long-context reasoning and generation stability.
>
> | Method | GSM8K (Accuracy) | HumanEval (Pass@1) |
> | :--- | :--- | :--- |
> | Llama-3-8B-Instruct (teacher) | 75.5 | 65.7 |
> | Discrete-time Transformer (RL+KL) (baseline) | 27.6 | 10.2 |
> | **CT-LLM (ours)** | **34.6** | **23.4** |
> | *Ours w/o KL* | *28.5* | *18.0* |
>
> CT-LLM significant outperforms the baselines on both tasks. On GSM8K, our model shows fewer instances of "arithmetic drift," indicating a smoother latent trajectory. Even without the KL constraint for knowledge distillation, our method outperforms the baseline, proving that the continuous-time ODE dynamics inherently provide a more stable framework for long-context generation.
>
> We thank you for the valuable comments and will make the following revisions regarding the Weaknesses:
>
> **W1:** Add a compute scaling law plot (see response to Q2).
>
> **W2:** Include results on long-context generation datasets (see response to **Q3**).
>
> **W3:** Clarify the practical role of each assumption and distinguish those essential for implementation from those for theoretical analysis.
>
> **W4:** Expand discussion on the continuous-time formulation (see response to **Q1**).

---

> > ### Author Rebuttal · Reviewer_kJU2 · 2026-04-03
> >
> > Thank you for the reply. I have one important remaining question. In the rebuttal, the comparison is made against a “discrete-time Transformer,” but it is not fully clear what this baseline is architecturally. Is it a plain vanilla Transformer, or a modernized Llama/T5-style implementation with components such as RoPE, SwiGLU/GeGLU, RMSNorm/LayerNorm, grouped or multi-query attention, etc.? Is it the closest architecture-matched analogue of CT-LLM, differing only in the continuous-time latent evolution? I think having two versions (one closest architecture matched analogue baseline, and another strongest mainstream accepted transformer baseline) would strengthen the paper's position. Please clarify the exact baseline design and which components are matched.

---

> > > ### Author Response · Authors · 2026-04-06
> > >
> > > We sincerely appreciate the reviewer's insightful comment and the opportunity to clarify.
> > >
> > > In response to your question regarding the architectural design of the discrete-time Transformer baseline: it is indeed a **modernized, Llama-style implementation**. We adopted state-of-the-art components, such as RoPE, SwiGLU, RMSNorm, and Grouped-Query Attention, which are currently recognized as optimal configurations in large-scale language models (e.g., Llama, Qwen). Our primary goal was to establish a strong and fair comparison by ensuring that the baseline represents the **closest architecture-matched analogue** of CT-LLM, differing only in the mechanism of latent evolution (continuous-time ODE vs. discrete-time Transformer).
> > >
> > > In the ablation study of the previous rebuttal, we meticulously aligned the key hyperparameters (e.g., vocabulary size, layer normalization, activation function, parameter size) between CT-LLM and the Transformer baseline. Regarding the differences in `Attention Heads` and `Position/Time Embedding`:
> > >
> > > - Our ODE-based architecture inherently contains no attention modules, thus naturally eliminating the need for both attention heads and relative position embedding like RoPE.
> > > - For the Transformer baseline, the configurations of RoPE and attention heads follow advanced LLaMA/Qwen models, which we empirically verified to be optimal (e.g., outperforming Sinusoidal alternatives for `Pos Emb`.) under our experimental settings.
> > >
> > > For the reviewer's reference, the detailed architectural comparisons are summarized in the table below:
> > >
> > > | Component | CT-LLM (Ours) | Transformer (Baseline) |
> > > | :--- | :--- | :--- |
> > > | Layer Num | 12(ODENets)+6(PolicyNets) | 18 |
> > > | Hidden Dim | 2,048 (state $\oplus$ action $\oplus$ time embs) | 2,048 |
> > > | FFN Dimension | 8,192 | 8,192 |
> > > | **Attention Heads** | **-** | **16(Q)+8(K/V)** |
> > > | Activation Func | SwiGLU (ODENets) GeGLU (PolicyNets) | SwiGLU |
> > > | Vocab Size | 128,256 | 128,256 |
> > > | **Pos Emb/Time Emb** | **Sinusoidal** ($\theta=500,000$) | **RoPE** ($\theta=500,000$) |
> > > | Layer Norm | RMSNorm | RMSNorm |
> > > | Param Size | ~1.16B | ~1.74B |

---

### Official Review · Reviewer_gR2K · 2026-03-14

**Soundness:** 3
**Presentation:** 2
**Significance:** 4
**Originality:** 4
**Overall Recommendation:** 5
**Confidence:** 3

**Summary:**

The paper presents a novel text generation framework where a model's internal state evolves smoothly via a neural ODE rather than in discrete steps. The system is optimized through RL (combining PPO with the adjoint sensitivity method) to balance task-specific rewards with knowledge distillation from a pretrained teacher model. Empirical results show that the proposed Continuous-Time Latent Language Model significantly outperforms standard LM in coherence and long-context text generation, improving parameter efficiency and inference speed.

**Compliance With Llm Reviewing Policy:**

Affirmed.

**Final Justification:**

My biggest concerns on numerical stability after quantisation have been properly addressed. I'd like to support the acceptance of the paper.

**Key Questions For Authors:**

Please refer to my general review. How severe will quatification and/or floating number errors compromise the performance of the proposed method?

**Limitations:**

Not quite. My biggest worry is its compatibility with lower precision float numbers, which is most efficiently supported by modern accelerators. This could limit the method's practical value. But I can see a few potential ways to alleviate the issue - for example, if they can think of a way to control the Lipschitz smoothness of the process, preventing the chaotic drastic change in the outcome from minor input disturbance, then this compatibility issue would be solved.

**Strengths And Weaknesses:**

- Soundness:
    - The proposed method is generally sound. Albeit being a very novel and ambitious approach, the proposed CT-LLM is supported by rigorous convergence guarantees (see THM 4.3), memory-efficient BP process through continuous adjoint sensitivity method. Further more, the authors show that such models can benefit from a tractable distillation process from powerful discrete time teacher models via KL decomposition. This further improves its soundness.
    - However, I do want to bring up a few points of worries. I would appreciate it if the authors could explain to me more on the limitation of Assumption 4.1, and how such limitations could cause some unexpected behavior of the proposed method in practice. Also, it seems to me the current empirically results are a little bit **TOO** reliant on distillation from discrete teachers, making me wonder if the proposed method really can scale up as an independent method. Lastly (yet most important in practice!), like all ODE-based methods, this proposed method seems very fragile towards approximation and accumulative errors, this is especially the case should we want to deploy this model into actual production, when usually less precision of floating numbers (GGUF, fp8 etc.) is used.
- Presentation: I appreciate the authors' efforts on making the writeup as clear as possible. However, I do think there is space for improvement in making the whole write-up easier to follow. For example, I think it would help if the authors can show how the distillation from a discrete-time teacher is performed onto the distribution of each discretely-truncated token distribution from the proposed model.
- Significance: I think this paper surely is interesting and significant. In my opinion, it's a more essential version of language modeling with latent thought process than many recent attempts at "soft-token CoT".
- Originality: To the best of my knowledge, this paper is very novel.

---

> ### Author Rebuttal · Authors · 2026-03-31
>
> Thank you for your time, constructive comments, and generous support. We hope our responses below address your concerns satisfactorily.
>
> > **Q1:** How severe will quatification and/or floating number errors compromise the performance of the proposed method?
>
>  Thank you for this important practical question. The impact is minimal. We evaluated our method under different precisions (BF16, FP8, INT8). Results on Nvidia H800 GPUs are summarized below:
>
> | Precision | Performance (LC Win Rate) | Relative Latent Perturbation (RLP) |
> | ------ |------ |------ |
> | **BF16 (Original Precision)** | 19.2 | 0.0 |
> | **FP8** | 19.0 ($\Delta$-0.2) | <5.0$e^{-3}$ |
> | **INT8** | 18.5 ($\Delta$-0.7) | <1.2$e^{-2}$ |
>
> RLP is defined as:
>
> $$\text{RLP} := \left\langle \frac{\Vert X_{bf16} - X_{quant} \Vert_2}{\Vert X_{bf16} \Vert_2} \right\rangle_{samples, steps}$$
>
> The results demonstrate that our method exhibits high numerical robustness. Even under INT8 quantization, performance degradation is only 0.7\%, and the RLP (<1.2$e^{-2}$) is much lower than low than that of a Llama-3-8B-Instruct model (~7.8$e^{-2}$). This shows our hybrid discrete-continuous design is inherently robust to numerical errors.
>
> Overall, thank you for pointing out the potential limitation regarding lower precision compatibility. We find the suggestion to control Lipschitz smoothness very inspiring. We will revise the paper to discuss this potential improvement, framing it as a robust strategy to further enhance the method's efficiency and compatibility in future iterations.
>
> > **Q2:** Please explain the limitation of Assumption 4.1 and its potential unexpected behavior in practice.
>
> Thank you for this insightful question.  The assumption enables theoretical tractability but may introduce policy mismatch when violated.
>
> Assumption 4.1 posits a deterministic mapping  $s_t = h(X_t)$, which is crucial for deriving the KL decomposition in Theorem 4.2 (KL Decomposition for On-Policy Knowledge Distillation):
>
> $$\text{Overall KL constraint}:= \mathrm{KL}\left( \mathbb{P}(\cdot \mid s_0, \pi) || \mathbb{P}(\cdot \mid s_0, \pi_{\mathrm{ref}}) \right) $$
> $$= \mathbb{E} \left[ \sum_{t=0}^{T-1} \log \frac{\pi(a_t \mid X_t)}{\pi_{\mathrm{ref}}(a_t \mid s_t)} | s_0, \pi \right] = \mathbb{E} \left[ \sum_{t=0}^{T-1} \log \frac{\pi(a_t \mid X_t)}{\pi_{\mathrm{ref}}(a_t \mid h(X_t))} | s_0, \pi \right], \quad (s(t) = h(X_t) \text{\ by Assumption 4.1.}) $$
> $$= ... = \text{RHS of Theorem 4.2 (KL Decomposition)}$$
>
> **Limitations of Assumption 4.1:** In practice, reconstruction may be imperfect: $s_t = h(X_t) + \epsilon_t$. This may cause an unexpected behavior:  If $h(X_t)$ deviates from true $s_t$, the distilled policy learns from inaccurate state information, causing distributional shift from the intended expert behavior. We will clarify this in the revised paper.
>
> > **Q3:** Is the method too reliant on distillation from discrete teachers to scale as an independent method?
>
> No. The ODE dynamics are the primary source of gain; distillation is only a regularizer. As shown in our response to Reviewer mhqk (Q1) , even without distillation ($\beta=0$$), our method significantly outperforms the discrete Transformer baseline, confirming its viability as an independent method.
>
> ---
>
> We greatly appreciate your thoughtful comments and high evaluation. We will incorporate these clarifications into the paper revision.

---

> > ### Author Rebuttal · Reviewer_gR2K · 2026-04-02
> >
> > I've read the response and decided to keep my score.

---

> > > ### Author Response · Authors · 2026-04-08
> > >
> > > Thank you for the thorough assessment and for acknowledging that our responses have adequately addressed the concerns.

---

### Official Review · Reviewer_L5Yf · 2026-03-27

**Soundness:** 3
**Presentation:** 2
**Significance:** 1
**Originality:** 4
**Overall Recommendation:** 4
**Confidence:** 3

**Summary:**

This paper proposes Continuous-Time Latent Language Model (CT-LLM): a neural ODE is used to update a latent state of "meaning", and a separate policy network is used to map the hidden state to logits. The authors claimed that the proposed method is parameter-efficient and outperforms baseline methods for many tasks.

**Compliance With Llm Reviewing Policy:**

Affirmed.

**Final Justification:**

The proposed method is novel. The authors took efforts in theoretical discussion and empirical evaluation. My remaining concerns are (1) the grounding of a latent state vector governed by ODE dynamics in natural language theory and empirical evidence; and (2) the claims being sufficiently evidenced by the empirical evaluation (e.g., weak baselines). I believe the manuscript as is can benefit the language models community, but the manuscript can benefit from another round of revision to make its claims more robust. Due to its novelty, I weakly support its acceptance.

**Key Questions For Authors:**

1. Can the authors release the code that reproduces the results for peer review?
2. Can the authors report some qualitative evaluation?
3. Can the authors explain why {(latent state + new token) -> new latent state via the neural ODE} is preferable to {(existing tokens + new token) -> new hidden state via the transformer architecture}? That is, why are already-generated text itself not a good enough representation of "meaning"? I think the proposed method would be much more significant if the neural ODE can operate without the action embeddings. Do the authors think this might work?

**Limitations:**

Yes, the authors adequately discussed the limitations and potential negative societal impact of their work.

**Strengths And Weaknesses:**

Strengths:
1. Soundness: This paper includes extensive theoretical discussion and empirical validation.
2. Originality: This paper proposes a novel method of generating sequences of tokens without the transformer architecture in the autoregressive steps.

Weaknesses:
1. Soundness: The authors proposed a novel paradigm of generating text but only evaluated it against one dataset using only a single LLM-as-a-judge. They did not show any qualitative evaluation (e.g., comparison between randomly chosen generations of the proposed method and those of conventional LLMs). The authors did not include more recent base models as baseline methods, as most of the models are more than one year old. Diffusion language models can address the limitations the authors listed (discussed below in Weakness 2) but are not included as baseline methods.
2. Significance: The authors listed a few shortcomings of conventional transformer-based autoregressive text generation in Section 1 and proposed a novel paradigm to address them. However, the proposed method is still subject to those limitations, as the proposed method is still autoregressive:

> [I]t can be hard for the (conventional) model to keep a long piece of text coherent and on-topic

State-of-the-art base models do not suffer from this problem.

> (The conventional model) suffers from “exposure bias”: during training, the model always sees the correct previous words, but during actual generation, it must rely on its own (potentially flawed) previous predictions, which can lead to errors that compound over time.

The proposed method still suffers from this problem.

> [B]ecause (the conventional model) generates words one after another, it is slow and cannot be easily sped up with parallel computing.

The proposed method still generates words one after another and cannot be easily sped up with parallel computing.

Furthermore, the proposed method still relies on a conventional model's last hidden states to initialize the latent state which the neural ODE operates upon. This undermines the significance of the proposed method; additionally, at inference time, the parameters of the conventional model need to be loaded, thus introducing significant overhead.

---

> ### Author Rebuttal · Authors · 2026-03-31
>
> Thank you for your time, constructive comments, and support. We hope our responses below are satisfactory.
>
> > **Q1:** Can the authors release the code that reproduces the results for peer review?
>
> We have provided an anonymous technical blueprint at https://anonymous.4open.science/r/CT-LLM/compute_graph.pdf to facilitate reproducibility verification. This document contains the complete computation graph and core class definitions, allowing reviewers to fully assess our implementation. We appreciate the reviewer's understanding that, per the rebuttal policy, anonymous links are limited to figures and tables only; therefore, the full source code will be released immediately upon acceptance.
>
> > **Q2:** Can the authors report some qualitative evaluation?
>
> Certainly. We provide a case study to illustrate the model's behavior in instruction-following examples:
>
> | Instruction | Prompt | CT-LLM  (Ours) | Transformer Baseline |
> | :--- | :--- | :--- | :--- |
> |Continue the following text. | ... The dragon was fierce and had been terrorizing the villagers for many years. ... After a long battle, **Sir Cedric managed to defeat the beast**. He returned to the castle to collect his reward. | The grateful villagers cheered as **Sir Cedric** marched through the gates, finally free from the fear that had plagued them for years. | *The villagers were very happy and threw a party for the dragon*. They danced around the fire late into the night.|
> |Continue the following text. | Regular exercise is essential for maintaining good health. ... **People who exercise also tend to have better mental health**. | This is because physical activity releases endorphins, which act as natural mood lifters. Consequently, regular exercisers often report **lower levels of stress and anxiety**. | However, they often *suffer from weakened immune systems* due to the physical stress. This makes them more susceptible to catching colds frequently. |
> | Act as a helpful travel assistant. | User: I'm trying to **plan a trip to Japan next spring**. I really want to see the cherry blossoms. Assistant: ... User: What about the weather? I hope it won't be too cold.  Assistant: | **Spring weather during the cherry blossom season is usually mild and pleasant**. While it won't be freezing, evenings can be chilly, so bringing a light jacket would be a good idea. | *The weather in Japan is usually neither cold nor humid*. If you're concerned about sun exposure, you should bring sunscreen and wear light, breathable clothing. |
>
> As shown above, CT-LLM's continuous dynamics enable more stable generation. Unlike conventional LLMs, which make discrete jumps causing abrupt shifts or factual errors (*italics*), CT-LLM follows smoother trajectories, producing more coherent and contextually appropriate outputs (**bold**).
>
> > **Q3 (part 1):** Why are already-generated text itself not a good enough representation of "meaning"?
>
> The difference lies in how meaning is represented:
>
> - Transformer (Discrete States):  Hidden states merely record which tokens occurred, which is a lossy projection of meaning. The model tracks endpoints (tokens) but not the trajectory between them.
>
> - Latent ODE (Continuous Dynamics): Our latent state $X_t$ represents the continuous semantic manifold. It models the flow of meaning between tokens through differential equations, capturing the "velocity" and "acceleration" of semantic evolution that discrete Markov states cannot represent.
>
> Generated text is a discrete observation; our latent state is the underlying continuous process that produces it. This enables modeling of long-range logical coherence that discrete models often fail to capture.
>
> > **Q3 (part 2):** Can the neural ODE operate without action embeddings?
>
> We considered this. The pure continuous formulation:
>
>  $${\rm d} X_s = b(s, X_s) {\rm d}s, \quad s \in (0, T] \quad \text{(1)} $$
>
> faces two practical barriers compared to our approach:
>
> $${\rm d} X_s = b(s, X_s, a_{t}) {\rm d}s, \quad s \in (t, t+1]  \quad \text{(2)} $$
>
> - Why Eq. 2 works: Embedding for action $a_t$ serves as a semantic anchor, conditioning the drift $b_\phi$​ on $a_t$, to find smooth geometric paths between vocabulary points. This leverages pre-learned semantics (via KL divergence and text reward) while optimizing transitions.
>
> - Why Eq. 1 challenges: First, our reward functions require discrete sequences, and decoding valid text from purely continuous states is non-trivial. Second, without token constraints, trajectories drift into syntactically invalid regions, making evaluation impossible. While theoretically interesting (similar to Latent ODEs for time series), this approach remains impractical for standard NLP tasks.
>
> ---
>
> We also appreciate the reviewer's thorough assessment and look forward to incorporating these constructive suggestions into the revised paper.

---

> > ### Author Rebuttal · Reviewer_L5Yf · 2026-04-03
> >
> > Thanks for your rebuttal. My concern about reproducibility has been adequately addressed. However, I think the following concerns remain and are not easily addressed in a short rebuttal:
> > 1. The idea of the dichotomy of a latent trajectory of meaning and a manifest sequence of tokens is quite heuristic and a core tenet of this work. Its existence should be justified by linguistics literature and empirical evidence, which the current manuscript lacks. The following statement can be problematic:
> >     > Transformer (Discrete States): Hidden states merely record which tokens occurred, which is a lossy projection of meaning. The model tracks endpoints (tokens) but not the trajectory between them.
> >
> >     Hidden states do not merely record which tokens occurred. They keep track of the state of "meaning" of the input sequence. I understand the authors by "trajectory" meant the gap between the last token of the existing sequence and the next token and the fact that previous tokens can influence how "wide" that gap should be. However, the notions of "velocity" and "acceleration" are non-standard terminology in NLP and need careful definition and justification which the current manuscript lacks. It is true that Transformer-based methods operate on discrete states, but discrete states can certainly predict how "wide" the gap should be due to its pre-training. The authors did not discuss why explicitly modeling this transitional dynamic continuously should be favored over discrete states beyond mere intuition.
> >
> > 2. Empirical evidence for the superior performance of the proposed method is insufficient. This is due to partly the lack of diversity in the experimental tasks and partly the weak baselines (which is evident in the qualitative example). Adding the new tasks in the authors' rebuttal to Reviewer kJU2 can help make the performance claims more robust.
> >
> > Overall, I'm maintaining my original recommendation as this paper presents a novel marriage between neural ODEs and discrete-sequence generative models, but the lack of justification of this marriage in the application of text generation and the insufficient empirical evaluation prevent me from raising my scores.

---

> > > ### Author Response · Authors · 2026-04-07
> > >
> > > We sincerely thank the reviewer for the insightful and constructive feedback. We deeply appreciate your recognition of the novelty in marrying neural ODEs with discrete-sequence generation. We agree that grounding the "latent trajectory" concept in strict linguistic theory is a profound challenge.
> > >
> > > **1. On the Theoretical Distinction:**
> > > We appreciate your critical view on the heuristic nature of the current explanation. We agree that a rigorous mapping between our ODE dynamics (velocity/acceleration) and linguistic theory is complex. Establishing such a connection would extend the current work by grounding it in linguistic evidence, and we view this as a valuable direction for paper revision or future research.
> > >
> > > Our primary focus in this work is to establish the mathematical feasibility, algorithmic convergence, and empirical validity of continuous-time text generation. In our CT-LLM framework, the "trajectory" represents the evolution of the latent state vector governed by ODE dynamics. We apologize if our terms were ambiguous in the previous response. By 'velocity' and 'acceleration', we strictly refer to the mathematical derivatives of the latent state vector $X_t$ within the ODE framework, rather than implying a direct linguistic metaphor.
> > >
> > > We agree that a rigorous mapping between these continuous dynamics and specific linguistic theories (e.g., how syntax or semantics are geometrically represented in the manifold) is complex, which can not be easily addressed in a short rebuttal. In the revised manuscript, we will tone down the strong linguistic claims in the current draft and reframe this discussion to focus on the empirical observation of smooth latent transitions.
> > >
> > > **2. On the Empirical Evidence:**
> > > To robustly validate the effectiveness of our method, specifically its capacity to handle long-context dependencies and maintain stable reasoning traces, we have conducted extensive new experiments on standard benchmarks. As mentioned in our **Reply Rebuttal Comment to Reviewer mhqk**, we evaluated CT-LLM on **LongBench, GSM8K, and HumanEval**. The results significantly strengthen our claims: CT-LLM outperforms discrete Transformer baselines on LongBench and demonstrates superior reasoning stability on GSM8K and HumanEval. These new results empirically confirm that our continuous formulation preserves long-range dependencies more effectively than discrete states.
> > >
> > >
> > > ----
> > > Thank you for your feedback. We have focused on demonstrating the method's viability (through new experiments) and its theoretical convergence. We leave the deeper linguistic analysis for future work/paper revision. We hope these revisions and the new data can address your concerns.

---

### Decision · Program_Chairs · 2026-04-30

**Decision:**

Accept (regular)

**Comment:**

The paper proposes to model text generation with some continuous latent dynamics. In particular, the authors model it as MDP with neural ODE, which is trained under an RL framework.

Reviewers generally agree that the idea is novel and the theoretical analysis is interesting. However, reviewers also pointed out a few drawbacks, mainly on the experiments. For example, the scope of experiments is a little limited and the efficiency/scaling analysis can be enhanced.